# 🖼️ PAPER2VIDEO: AUTOMATIC VIDEO GENERATION FROM SCIENTIFIC PAPERS

## ABSTRACT

Academic presentation videos have become an essential medium for research communication, yet producing them remains highly labor-intensive, often requiring hours of slide design, recording, and editing for a short 2 to 10 minutes video. Unlike natural video, presentation video generation involves distinctive challenges: long-context inputs from research papers, dense multi-modal information (text, figures, tables), and the need to coordinate multiple aligned channels such as slides, subtitles, speech, and human talker. To address these challenges, we introduce **Paper2Video**, the first benchmark of 101 research papers paired with author-created presentation videos, slides, and speaker metadata. We further design four tailored evaluation metrics—Meta Similarity, PresentArena, *PresentQuiz*, and *IP Memory*—to measure how videos convey the paper's information to the audience. Building on this foundation, we propose **PaperTalker**, the first multi-agent framework for academic presentation video generation. It integrates slide generation with effective layout refinement by a novel effective *Tree Search Visual Choice*, cursor grounding, subtitling, speech synthesis, and talking-head rendering, while parallelizing slide-wise generation for efficiency. Experiments on Paper2Video demonstrate that the presentation videos produced by our approach are more faithful and informative than existing baselines, establishing a practical step toward automated and ready-to-use academic video generation.

## 1 INTRODUCTION

Academic presentation videos are widely used in research communication, serving as a crucial and effective means to bridge researchers, as many conferences require them as an essential material for submission. However, the manual creation of such a video is highly labor-intensive, requiring slide design, subtitle writing, per-slide recording, and careful editing, which on average may take several hours to produce a 2 to 10 minute video for a scientific paper. Despite some prior works on slide and poster generation [29, 38, 22] and other AI4Research tasks [4, 3, 12, 25, 20], automatic academic presentation video generation is a superproblem of them, a practical yet more challenging direction.

Unlike natural video generation [2, 37, 31, 10], presentation video exhibits distinctive characteristics, including multi-sensory integration, multi-figure conditioning, and high text density, which highlight the limitations of current natural video generation models [19]. Specifically, academic presentation video generation faces several crucial challenges: *a.* It originates from long-context papers that contain dense text as well as multiple figures and tables; *b.* It requires the coordination of multiple aligned channels, including slide generation [38], subtitling, text-to-speech [6], cursor control, and talking head generation [32, 8]; *c.* It lacks well-defined evaluation metrics: what constitutes a good presentation video, particularly in terms of knowledge conveyance and audience accessibility. Even for the state-of-the-art end-to-end video–audio generation model Veo3 [10], notable limitations remain in video length, clarity of dense on-screen text, and multi-modal long-document condition. In this work, we try to solve these two core problems as shown in Figure 1.

To enable comprehensive evaluation of academic presentation video generation, we present the **Paper2Video** Benchmark, comprising 101 paired research papers and author-recorded presentation videos from recent conferences, together with original slides and speaker identity metadata. Based on this benchmark, we develop a suite of metrics to comprehensively evaluate generation quality from multiple dimensions: **(i)** Meta Similarity — We employ a VLM to evaluate the alignment of generated slides and subtitles with human-designed counterparts. **(ii)** PresentArena — We use a VideoLLM as a proxy audience to perform double-order pairwise comparisons between generated and human-made videos. Notably, the primary purpose of a presentation is to *effectively convey the information contained in the paper*. To this end, we introduce **(iii) PresentQuiz**, which treats the VideoLLMs as the audience and requires them to answer paper-derived questions given the videos.

Figure 1: This work solves two core problems for academic presentations: **Left:** *how to create a presentation video from a paper?* PaperTalker – an agent integrates slide, subtitling, cursor grounding, speech synthesis, and talking-head video rendering. **Right:** *how to evaluate a presentation video?* Paper2Video – a benchmark with well-designed metrics to evaluate presentation quality.

Furthermore, another important purpose of presentation video is to *enhance the visibility and impact of the author's work*. Motivated by real-conference interactions, we introduce **(iv) IP Memory**, which measures how well an audience can associate authors and works after watching presentation videos.

To effectively generate ready-to-use academic presentation videos, we propose **PaperTalker**, the first multi-agent framework that enables academic presentation video generation from research papers and speaker identity. It integrates subsequent key modules: **(i)** Slide Generation. Instead of adopting the commonly used format (*e.g.*, pptx, XML) from a template slide as in [38], we employ LaTeX code for slide generation from sketch, given its formal suitability for academic use and higher efficiency. Specifically, we employ a state-of-the-art Coder to generate code and introduce an effective **focused debugging** strategy, which iteratively narrows the scope and resolves compilation errors using feedback that indicates the relevant rows. To address the insensitivity of LLMs to fine-grained numerical adjustments, we propose a novel method called **Tree Search Visual Choice**. This approach systematically explores parameter variations to generate multiple branches, which are then concatenated into a single figure. A VLM is then tasked with selecting the optimal branch, thereby effectively improving element layouts such as figure and font size. **(ii)** Subtitling and Cursor Grounding. We generate subtitles and cursor prompts for each sentence based on the slides. Then we achieve cursor spatial-temporal alignment using **Computer-use grounding model** [17, 23] models and WhisperX [1] respectively. **(iii)** Speech Synthesis and Talking-head Rendering. We synthesize personalized speech via text-to-speech models [5] and produce talking-head videos [8, 32] for author presentations. Inspired by human recording practice and the independence between each slide, we **parallelize generation** across slides, achieving a speedup of more than **6×**. We will open-source all our data and codebase to empower the research community.

To summarize, our contributions are as follows:

- We present Paper2Video, the first high-quality benchmark of 101 papers with author-recorded presentation videos, slides, and speaker metadata, together with evaluation metrics: Meta Similarity, PresentArena, PresentQuiz, and IP Memory.

- We propose PaperTalker, the first multi-agent framework for academic presentation video generation. It introduces three key modules: **(i)** tree search visual choice for fine-grained slide generation; **(ii)** a GUI-grounding model coupled with WhisperX for spatial-temporal aligned cursor grounding; and **(iii)** slide-wise parallel generation to improve efficiency.

- Results on Paper2Video confirm the effectiveness of PaperTalker, which outperforms human-made presentations by 10% in PresentQuiz accuracy and achieves comparable ratings in user studies, indicating that its quality approaches that of human-created content.

## 2 RELATED WORKS

### 2.1 VIDEO GENERATION

Recent advances in video diffusion models [2, 31, 14, 15] have substantially improved *natural* video generation in terms of length, quality, and controllability. However, these **end-to-end** diffusion models still struggle to produce long videos [10, 32] (*e.g.*, several minutes), handle multiple shots, and support conditioning on multiple images [19]. Moreover, most existing approaches generate only video without aligned audio, leaving a gap for real-world applications. To address these limitations, recent works leverage **multi-agent** collaboration to generate multi-shot, long video–audio pairs and enable multi-image conditioning. Specifically, for natural videos, MovieAgent [35] adopts a hierarchical CoT planning strategy and leverages LLMs to simulate the roles of a director, screenwriter, storyboard artist, and location manager, thereby enabling long-form movie generation. Alternatively, PresentAgent [26] targets presentation video generation but merely combines PPTAgent [38] with text-to-speech to produce narrated slides. However, it lacks personalization (*e.g.*, mechanical speech

Table 1: **Comparison of Paper2Video with existing benchmarks.** Top: existing natural video generation; Button: recent Agents for research works.

| Benchmarks | Inputs | Outputs | Subtitle | Slides | Cursor | Speaker | |
|---|---|---|---|---|---|---|---|
| | | | | | | Face | Voice |
| *Natural Video Generation* | | | | | | | |
| VBench [14] | Text | Short Vid. | ✗ | ✗ | ✗ | ✗ | ✗ |
| VBench++ [15] | Text&Image | Short Vid. | ✗ | ✗ | ✗ | ✗ | ✗ |
| Talkinghead [30] | Audio&Image | Short Vid. | ✗ | ✗ | ✗ | ✓ | ✓ |
| MovieBench [34] | Text&Audio&Image | Long Vid. | ✓ | ✗ | ✗ | ✓ | ✓ |
| *Multimodal Agent for Research* | | | | | | | |
| Paper2Poster [22] | Paper | Poster | ✗ | ✗ | ✗ | ✗ | ✗ |
| PPTAgent [38] | Doc.&Template | Slide | ✗ | ✓ | ✗ | ✗ | ✗ |
| PresentAgent [26] | Doc.&Template | Audio&Long Vid. | ✓ | ✓ | ✗ | ✗ | ✓ |
| Paper2Video (Ours) | Paper&Image&Audio | Audio&Long Vid. | ✓ | ✓ | ✓ | ✓ | ✓ |

and absence of a presenter) and fails to generate academic-style slides (*e.g.*, missing opening and outline slides), thereby limiting its applicability in academic contexts. Our work addresses these limitations and enables ready-to-use academic presentation video generation.

## 2.2 AI FOR RESEARCH

Many useful tasks have been explored under the umbrella of AI for Research (AI4Research) [4], which aims to support the full scholarly workflow spanning text [9], static visuals [22], and dynamic video [26]. With the breakthrough of LLMs in text generation and the Internet search ability, extensive efforts have been devoted to academic writing [3] and literature surveying [16, 11, 18, 12], substantially improving research efficiency. Besides, some works [28, 36] benchmark AI agents' end-to-end ability to replicate top-performing ML papers, while others leverage agents to enable idea proposal [27] and data-driven scientific inspiration [7, 21]. To further enhance productivity, a growing number of work focuses on the automatic visual design of figures [33], slides [38], posters [22], and charts [13]. More recently, Paper2Agent [20] has reimagined research papers as interactive and reliable AI agents, designed to assist readers in understanding scientific works. However, very few studies have investigated video generation for scientific purposes, leaving this area relatively underexplored. Our work belongs to one of the pioneering efforts in this direction, initiating a systematic study on academic presentation video generation.

## 3 PAPER2VIDEO BENCHMARK

### 3.1 TASK DEFINITION

Given a research paper and the author's identity information, our goal is to automatically synthesize an academic presentation video that faithfully conveys the paper's core contributions in an audience-friendly manner. We identify that a perfect presentation video is usually required to integrate four coordinated components: (***i***) **slides** contain well-organized, visually oriented, expressive figures and tables with concise text description; (***ii***) **synchronized subtitles and speech** are semantically aligned with the slides, including supplementary details; (***iii***) **presenter** should exhibit natural yet professional facial expressions, ideally accompanied by appropriate gestures; and (***iv***) **a cursor indicator** serves as an attentional anchor, helping the audience focus and follow the narration.

This task poses several distinctive challenges: ***a*. Multi-modal Long-Context Understanding.** Research papers span many pages with dense text, equations, figures, and tables. ***b*. Multi-turn Agent Tasks.** It is challenging to solve this task with a single end-to-end model, as it requires multi-channel generation and alignment (*e.g.*, slides, cursors, and presenter). ***c*. Personalized Presenter Synthesis.** Achieving high-quality, identity-preserving, and lip-synchronous talking-head video remains time-consuming, and even more challenging when jointly modeling voice, face, and gesture. ***d*. Spatial-Temporal-Grounding.** Producing cursor trajectories synchronized with narration and slide content demands precise alignment between linguistic units and visual anchors.

### 3.2 DATA CURATION

**Data Source.** We use AI conference papers as the data source for two reasons: (i) they offer high-quality, diverse content across subfields with rich text, figures, and tables; and (ii) the field's rapid growth and open-sharing culture provide plentiful, polished author-recorded presentations and slides on YouTube and SlidesLive. However, complete metadata are often unavailable (*e.g.*, presentation videos, slides, presenter images, and voice samples). We thus manually select papers

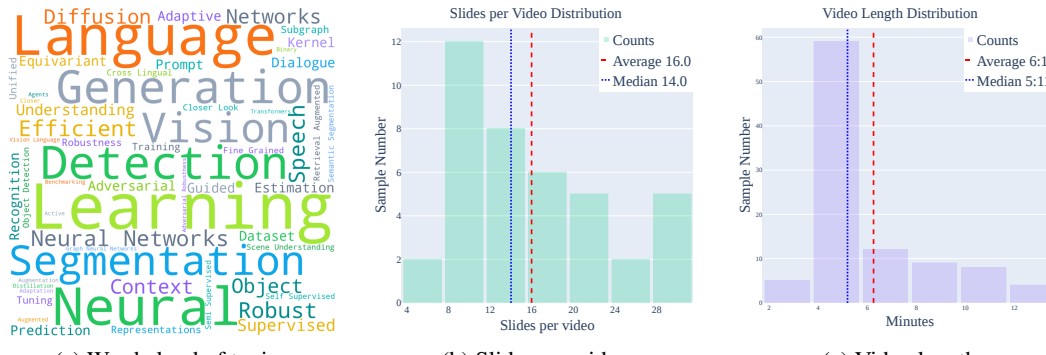

(a) Word cloud of topics      (b) Slides per video      (c) Video length

Figure 2: **Statistics of Paper2Video benchmark.** It spans diverse topics, with presentations comprising 4–28 slides and lasting 2–14 min, providing a valuable benchmark for the automatic generation and evaluation of academic presentation videos.

with relatively complete metadata and supplement missing fields by sourcing presenter images from authors' websites. Overall, we curate 101 peer-reviewed conference papers from the past three years: 41 from machine learning (*e.g.*, NeurIPS, ICLR, ICML), 40 from computer vision (*e.g.*, CVPR, ICCV, ECCV), and 20 from natural language processing(*e.g.*, ACL, EMNLP, NAACL). Each instance includes the paper's full LaTeX project and a matched, author-recorded presentation video comprising the slide and talking-head streams with speaker identity (*e.g.*, portrait and voice sample). For 40% of the data, we additionally collect the original slide files (PDF), enabling direct, reference-based evaluation of slide generation.

**Data Statistics.** Overall, Paper2Video covers 101 paper-video pairs spanning diverse topics as shown in Figure 2 (a), ensuring broad coverage across fields. The paper contains $13.3K$ words($3.3K$ tokens), 44.7 figures, and 28.7 pages on average, serving as multi-modal long document inputs. As illustrated in Figure 2 (b) and (c), we also report the distributions of slides per presentation and video durations in Paper2Video. On average, presentations contain 16 slides and last 6min 15s, with some samples reaching up to 14 minutes. Although Paper2Video comprises 101 curated presentations, the benchmark is designed to evaluate long-horizon agentic tasks rather than mere video generation.

### 3.3 EVALUATION METRICS

Unlike natural video generation, academic presentation videos serve a highly specialized role: they are not merely about visual fidelity but about communicating scholarship. This makes it difficult to directly apply conventional metrics from video synthesis (*e.g.*, FVD, IS, or CLIP-based similarity). Instead, their value lies in how well they *disseminate research, amplify scholarly visibility.*

From this perspective, we argue that a high-quality academic presentation video should be judged along two complementary dimensions (see Figure 3): **For the audience:** the video is expected to faithfully convey the paper's core ideas(*i.e.,* motivation and contributions), while remaining accessible to audiences. **For the author:** the video should foreground the authors' intellectual contribution and identity, and enhance the work's visibility and impact. To systematically capture these goals, we introduce tailored evaluation metrics specifically designed for academic presentation videos.

**Meta Similarity** – *How video like human-made?* As we have the ground-truth human-made presentation videos with original slides, we evaluate how well the generated intermediate assets (*i.e.,* speech, slides, and subtitles) aligned with the ones created by authors, which serves as the pseudo ground-truth. (i) For each slide, we pair the slide image with its corresponding subtitles and submit both the generated pair and the human-made pair to the VLMs to obtain a similarity score on a five-point scale. (ii) To further assess speech(*i.e.,* vocal timbre), we uniformly sample a ten-second segment from the presentation audio, encode the generated and human-recorded audio with a speaking embedding model [24], and compute the cosine similarity between the embeddings to measure speech similarity.

**PresentArena** – *Which video is better?* Similar to the human audience watching the presentation, we employ the VideoLLMs as the proxy audience to conduct pairwise comparisons of presentation videos, where the winning rate serves as the metric. For each pair, the model is queried twice in opposite orders: $(A, B)$ and $(B, A)$. This procedure reduces hallucinations and position bias. The two judgments are then aggregated by averaging to obtain a more stable preference estimation.

**PresentQuiz** – *How videos conveys the paper knowledge?* Following prior work [22], we evaluate information coverage using a multiple-choice quiz on the presentation video. We first generate a set

Figure 3: **Overview of evaluation metrics.** We propose three metrics that systematically evaluate academic presentation video generation from the perspective of the relationship between the generated video and **(i)** the original paper and **(ii)** the human-made video.

of questions with four options and the corresponding correct answers from the source paper. Then we ask the VideoLLMs to watch the presentation and answer each question. Overall accuracy serves as the metric, with higher accuracy indicating better information coverage.

**IP Memory** – *How videos affect the author's visibility and work impact?* Another key purpose of academic presentation videos is to enhance the visibility and impact of the author's work. Yet, this metric is unclear and difficult to simulate and thus remains an open problem. In real-conference settings, audiences who recall a scholar after attending their presentation are more inclined to pose relevant questions in later interactions. Motivated by this phenomenon, we propose a metric to assess how effectively a presentation video enables the audience to recall the work. Additional implementation details are provided in Appendix B.1.

Furthermore, to ablate the contribution of each component, we evaluate both the quality and the gains provided by individual components (*e.g.,* slides, cursor, and presenter). Notably, to further assess presentation videos from the user perspective, we conduct human studies to evaluate the results.

## 4 PAPERTALKER AGENT

**Overview.** To address these challenges and liberate researchers from the burdensome task of manual video preparation, we introduce PaperTalker, a multi-agent framework designed to automatically generate presentation videos directly from academic papers. As illustrated in Figure 4, to decouple the different roles, making the method scalable and flexible, the pipeline comprises four builders: **(i)** Slide builder. Given the paper, we first synthesize slides with LATEX code and refine them with compilation feedback to correct grammar and optimize layout; **(ii)** Subtitle builder. The slides are then processed by a VLM to generate subtitles and sentence-level visual-focus prompts; **(iii)** Cursor builder. These prompts are then grounded into on-screen cursor coordinates and synchronized with the narration. **(iv)** Talker builder. Given the voice sample and the portrait of the speaker, text-to-speech and talking-head modules generate a realistic, personalized talker video. For clarity, we denote the paper document, author portrait, and voice sample as $\mathcal{D}$, $\mathcal{I}$, and $\mathcal{A}$, respectively.

### 4.1 SLIDE BUILDER

A prerequisite for producing a presentation video is the creation of the slides. Despite there being some existing works [38], we target the generation of academic slides with fine-grained layouts and formal structure from scratch. Rather than selecting a template and iteratively editing it with VLMs, we generate slides directly from a paper's LATEX project by prompting the model to write Beamer code. We adopt Beamer for three reasons: **(i)** LATEX's declarative typesetting automatically arranges text block and figures from their parameters without explicitly planing the positions; **(ii)** Beamer is compact and expressive, representing the same content in fewer lines than XML-based formats; and **(iii)** Beamer provides well-designed, formally configured styles (*e.g.*, page numbers, section headers, hyperlinks) that are well suited to academic slide design.

Given the paper $\mathcal{D}$ as input, the LLM first produces a draft slide code. We compile this code to collect diagnostics(*i.e.,* errors and warnings). Then, we use the error information to elicit a repaired correct code. This procedure ensures that the generated Beamer code is grammatically correct and effectively leverages and faithfully covers its content.

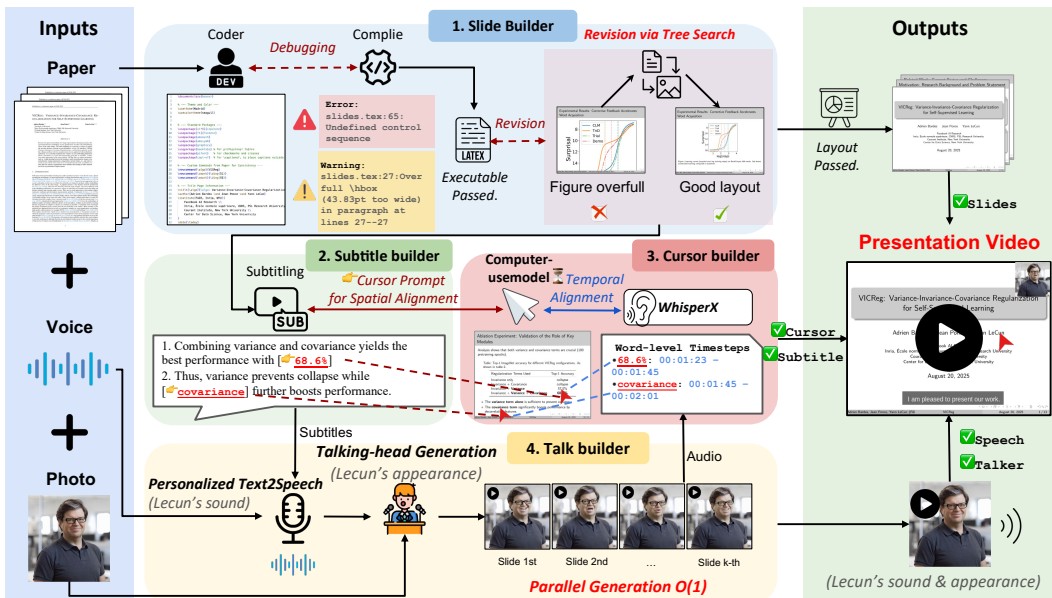

Figure 4: **Overview of PaperTalker.** Our pipeline comprises three key modules: **(i)** tree search visual choice for fine-grained slide layout optimization; **(ii)** a GUI-grounded model paired with WhisperX for spatiotemporally aligned cursor grounding; and **(iii)** slide-wise parallel generation for efficiency.

Although LaTeX can automatically arrange the location of the contents in the slides, the generated slides could sometimes still suffer from inappropriate layouts (*e.g.*, overflow) due to the unsuitable parameters for figure or text font size. However, as the compilation warning signals potential layout issues, we are able to first use them to identify the slides that require refinement.

**Tree Search Visual Choice.** After localizing the slides that require refinement, the key challenge is how to adjust their layouts effectively. As LLMs/VLMs fail to perceive real-time visual feedback like human designers, we observe that prompting the them to directly tune numeric layout parameters (*e.g.*, font sizes, margins, figure scales) is ineffective: the models are largely **insensitive** to small numeric changes, yielding unstable and inefficient refinement, consistent with limitations of the parameter-editing strategy in PPTAgent [38]. To address this limitation, we introduce a *visual-selection* module for overflowed slides. The module first constructs the neighborhoods of layout variants for the current slide by rule-based adjusting the figure and text parameters, renders each variant to an image, and then uses the VLMs as a judge to score the candidates and select the one with the best layout. Specifically, for text-only slides, we sweep the font size; for slides with

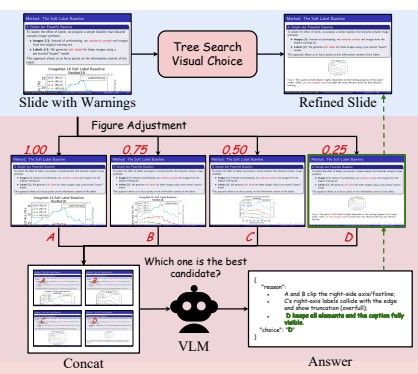

Figure 5: **Tree Search Visual Choice.** It combines a rule-based proposal mechanism with VLM-based scoring to select the optimal candidate.

figures, we first vary the figure scaling factors (*e.g.*, 1.25, 0.75, 0.5, 0.25) and then reduce the font size, details shown in Figure 5. These edits are straightforward in LaTeX Beamer, whose structured syntax automatically reflows content as parameter changes. This module **decouples discrete layout search from semantic reasoning** and reliably resolves overflow cases with minimal time and tokens.

After fixing the errors and adjusting the parameters, we compile the slide code to obtain the finalized slides $\mathcal{S}_i, i = 1, \ldots, n$ with fine-grained layouts, where $n$ indicates the number of slides.

## 4.2 SUBTITLE BUILDER

As the speech should follow the slides, given the generated slide $\mathcal{S}_i$, we rasterize them into images and pass them to a VLM, which produces sentence-level subtitles $T_i^j$ and its corresponding visual-focus prompt $P_i^j$. The visual-focus prompt serves as an intermediate representation linking speech to the cursor, enabling precise temporal and spatial alignment of the cursor with the narration in order to improve audience guidance, which will be discussed in Section 4.4.

### 4.3 TALKER BUILDER

The presenter video is vital for audience engagement and conveying the researcher's scholarly identity (*e.g.*, face and voice). Given the subtitles $\mathcal{T}_i$, the author's portrait $\mathcal{I}$, and a short voice sample $\mathcal{A}$, our objective is to synthesize a presenter video that delivers the slide content in the author's voice, with faithful identity preservation and lip–audio synchronization.

**Subtitle-to-Speech.** Given subtitles and voice sample, we use F5-TTS [6] to generate speech audio per slide, $\widetilde{\mathcal{A}}_i = \mathrm{TTS}\big(\{T_i^j\}_{j=1}^{m_i}|\mathcal{A}\big), i = 1, \ldots, n,$ where $m_i$ is the number of the sentences in $\mathcal{T}_i$.

**Parallel Talkinghead Generation.** To balance fidelity and efficiency, we use Hallo2 [8] for head-only synthesis and employ FantasyTalking [32] to support talking generation with upper-body articulation. A persistent challenge is the long generation time: generating only a few minutes of talking-head video typically takes several hours, and some models(*e.g.*, FantasyTalking) do not yet natively support long-video generation. Inspired by the common practice of slide-by-slide recording and the independence between each slide, we synthesize the presenter video on a per-slide basis. Specifically, for each slide $\mathcal{S}_i$, given the audio condition $\widetilde{\mathcal{A}}_i$ and portrait $\mathcal{I}$, we generate an independent clip $\mathcal{V}_i$ and execute these jobs in parallel, markedly reducing generation time: $\mathcal{V}_i = \mathcal{G}\big(\widetilde{\mathcal{A}}_i, \mathcal{I}\big), i = 1, \ldots, n,$ where $\mathcal{G}$ represents the talking-head generation model. This design is justified because slide transitions are hard scene changes, and the temporal continuity of the presenter across adjacent slides is unnecessary.

### 4.4 CURSOR BUILDER

**Spatial-Temporal Grounding.** In practice, presenters leverage the cursor as an attentional guide: a well-aligned cursor trajectory minimizes extraneous cognitive load, helps the audience track the presentation, and keeps focus on the key content. However, automatic cursor-trajectory grounding is nontrivial, requiring simultaneous alignment to the timing of speech and the visual semantics of the slides. To simplify the task, we assume that the cursor will stay still within a sentence and only move between the sentences. Thus, we estimate a per-sentence cursor location and time span. For spatial alignment, motivated by strong computer-use models [17, 23] which simulate user interaction with the screenshot, we propose to ground the cursor location $(x, y)$ for each sentence with the visual focus prompt $\mathcal{P}_i^j$ by UI-TARS [23]. To achieve precise temporal alignment, we then use WhisperX [1] to extract word-level timestamps and align them with the corresponding sentence in the subtitles to derive the start and end times $(t_s, t_e)$ of each cursor segment.

## 5 EXPERIMENTS

### 5.1 BASELINE AND SETTINGS

We evaluate three categories of baselines: **(i)** End-to-end Methods [31, 10], where natural video generation models produce the presentation video directly from a prompt generated by paper; **(ii)** Multi-Agent Frameworks [26, 38], which combine slide generation with text-to-speech generation and compose them into a presentation video; and **(iii)** PaperTalker, our method and its variants. For the VLM and VideoLLM, we choose *GPT-4.1* and *Gemini-2.5-Flash*, respectively, for a favorable efficiency and performance trade-off. We perform inference using eight NVIDIA RTX A6000 GPUs.

### 5.2 MAIN RESULTS

**Meta Similarity.** We evaluate the alignment of the generated slides, subtitles, and speech with corresponding human-authored ones. For speech, we randomly sample a 10-second audio segment from the video generated by each method and compute the cosine similarity between its embeddings [24] and those of the author's speech. As shown in Table 2, PaperTalker attains the highest scores in both speech and content similarity, demonstrating that its outputs **align most closely with human creation** among all baselines. We attribute this performance to personalized TTS and our slide-generation design: **(i)** adopting *Beamer*, which provides formal, academically styled templates while LaTeX automatically arranges content within each slide; and **(ii)** a tree search visual choice layout refinement that further enforces fine-grained slide layouts as commonly observed in human-authored slides.

**PresentArena.** We compare the presentation videos generated by each method against the human-made videos. As an automatic evaluator, we prompt the VideoLLMs as a judge to determine which presentation is better with respect to clarity, delivery, and engagement. As shown in Table 2, PaperTalker attains the highest pairwise winning rate among all baselines, indicating that our method produces presentation videos with **superior overall perceived quality**. Notably, PaperTalker outperforms its variants without the talker and cursor by 1.8%, highlighting the gains introduced by these components and implying that the VideoLLM **favors presentation videos with a talker presenting**.

**PresentQuiz.** To assess information coverage, we conduct a VideoQA evaluation. Following prior work on posters [22], we construct QA sets by prompting an LLM to generate questions targeting

Table 2: **Detailed evaluation result of Paper2Video across three baselines.** PaperTalk* represents a simple version without presenter and cursor. **Bold** and Underline indicates the best and the second.

| Method | Similarity↑ | | Arena↑ | PresentQuiz Acc.↑ | | IP Memory↑ | Avg. Duration(s) |
|---|---|---|---|---|---|---|---|
| | Speech | Content | | Detail | Under. | | |
| HumanMade | **1.00** | **5.00** | **50.0%** | 0.738 | 0.908 | **52.4%** | 375.15 |
| Wan2.2 [31] | NA | NA | 1.1% | 0.251 | 0.551 | 11.5% | 4.00 |
| Veo3 [10] | 0.133 | NA | 1.2% | 0.367 | 0.585 | 31.3% | 8.00 |
| PresentAgent_QWEN [26] | 0.045 | 0.24 | 1.2% | 0.404 | 0.625 | 10.4% | 494.06 |
| PresentAgent_GPT4.1 [26] | 0.045 | 1.47 | 2.0% | 0.548 | 0.654 | 12.5% | 430.20 |
| PaperTalk_QWEN | 0.646 | 1.66 | 10.8% | 0.635 | 0.863 | 39.5% | 190.05 |
| PaperTalk_GPT4.1* | 0.646 | 1.97 | 15.2% | 0.835 | 0.949 | 37.5% | 234.36 |
| PaperTalk_GPT4.1 | 0.646 | 1.97 | 17.0% | **0.842** | **0.951** | **50.0%** | 234.36 |

(i) fine-grained details and (ii) higher-level understanding of the paper. The videos and QA sets are then fed into a VideoLLM to conduct the quiz. As shown in Table 2, PaperTalker achieves superior performance across both aspects, outperforming HumanMade and PresentAgent despite shorter video length. This indicates that PaperTalker produces videos that are **more informative within shorter durations**. Furthermore, the absence of the talker or cursor results in performance degradation, as the cursor trajectory potentially **guides the attention and supports accurate grounding of the key contents** for the VideoLLMs during inference, referring to Table 4 for more details.

**IP Memory.** We evaluate the degree to which the generated presentation videos facilitate audience retention of the work, thereby assessing their memorability and lasting impact. PaperTalker achieves the highest recall accuracy. This improvement mainly stems from the inclusion of an engaging talker with the author's figure and voice, which **significantly helps the audience retain the video content**.

**Human Evaluation.** To further assess the quality of the generated presentations from the user perspective, we conducted a human evaluation in which twenty-five participants were provided with each paper along with its corresponding presentation videos generated by different methods. Participants were asked to rank the videos according to their preferences(1(worse) − 5(best)). As shown in Figure 6, human-made videos achieve the highest score, with PaperTalker ranking second and outperforming all other baselines. This demonstrates that presentation videos generated by PaperTalker **gain consistently favor from human users** over other baselines and **comparable to human-made**.

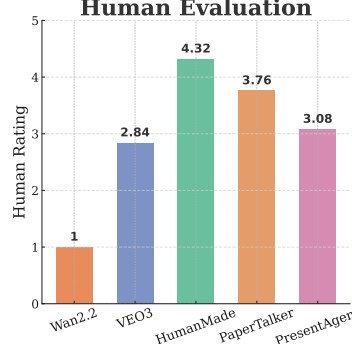

Figure 6: **Human evaluation**. We randomly sample the generated results from ten papers for evaluation.

**Efficiency Analysis.** As shown in Table 3, PaperTalker achieves **the lowest money cost** among all baselines. This efficiency arises from our slide builder design: leveraging *Beamer* reduces token consumption while the tree search visual choice layout refinement serves as a lightweight post-processing step. By contrast, PresentAgent incurs higher token costs due to frequent refinement queries for slide editing. As a result, the generation speed of PaperTalker without the talker is **more than twice as fast as** PresentAgent. Moreover, our parallel talking generation mechanism further reduces runtime **by nearly sixfold**.

## 5.3 QUALITATIVE ANALYSIS

As shown in Figure 7, PaperTalker produces presentation videos that **most closely align with the human-made** ones. While Veo3 [10] renders a high-quality speaker in front of the screen, it is constrained by short duration (*e.g.*, 8s) and blurred text

Table 3: **Generation cost for each method.**

| Method | Token (K) ↓ | Time (min.)↓ | Cost ($)↓ |
|---|---|---|---|
| Veo3 [10] | NA | **0.4** | 1.667 |
| Wan2.2 [31] | NA | 8.1 | 0.280 |
| PresentAgent [26] | 241 | 39.5 | 0.003 |
| PaperTalker (w/o Talker) | **62** | 15.6 | **0.001** |
| PaperTalker (w/o Par.) | **62** | 287.2 | **0.001** |
| PaperTalker | **62** | 48.1 (**6×**) | **0.001** |

(*e.g.*, "Labuel," "sofft"). Besides, PresentAgent[26] typically suffers from the absence of the presenter and slide-design errors (*e.g.*, overflow, incorrect title, incomplete author lists, and institutions). By contrast, PaperTalker produces the most human-like presentation videos with well-structured slides, a consistent presenter, and accurate cursor highlights. Its fine-grained slide quality is mainly attributed to two factors: **(i)** the LaTeX compiler **automatically optimizes** the placement of text and figures, leaving only high-level layout choices (*e.g.*, single or double columns, element sizes) to be specified;

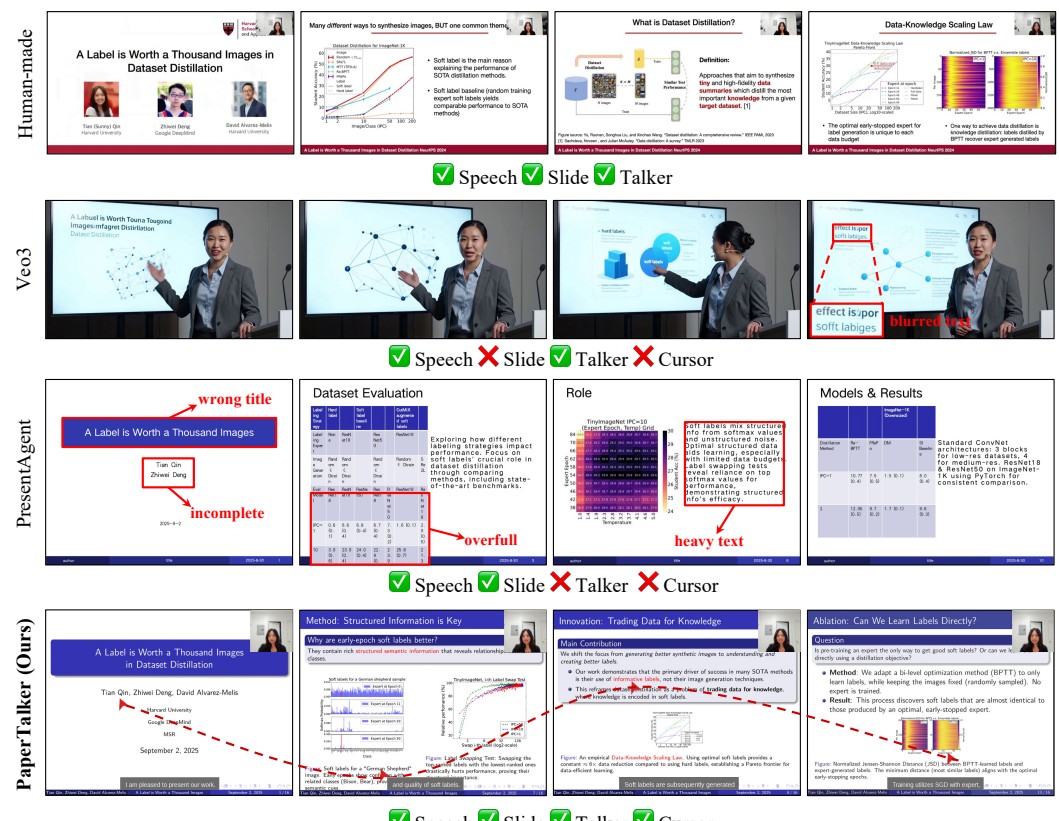

Figure 7: **Visualization of generated results.** PaperTalker produces presentation videos with rich, fine-grained slide content, accurate cursor grounding, and an engaging talker; in contrast, Veo3 [10] yields blurred text and incomplete information coverage, while PresentAgent [26] produces text-heavy slides and suffers from overfull layout issues and inaccurate information (*e.g.*, title and institutions).

**(ii)**, our Tree-Search Visual Choice module generates neighboring candidate parameters via rule-based heuristics and uses a VLM solely to **rank and select** among them. Since VLMs are insensitive to subtle numerical changes, **decoupling parameter adjustment from candidate selection** leads to more stable and reliable layout refinement. Additional visualizations are provided in Figure 8.

## 5.4 KEY ABLATIONS

**What benefits are brought by Cursor Highlight?** Motivated by the observation that a cursor typically helps audiences locate the relevant region, we hypothesize that a visible cursor, by providing an explicit spatial cue, facilitates content grounding for VLMs. To evaluate this, we design a localization QA task: for each subtitle sentence and its corresponding slide, a VLM generates a four-option multiple-choice question about the sentence's corresponding position on the slide. The VLMs are then prompted to answer using slide screenshots, with or without the cursor, and accuracy is measured as the metric. As shown in Table 4, the accuracy is much higher with the cursor highlight, corroborating its **importance for the audience's visual grounding accessibility** of presentation videos.

Table 4: **Ablation study on cursor.**

| Method | Accuracy↑ |
| --- | --- |
| PaperTalker (w/o Cursor) | 0.084 |
| PaperTalker | **0.633** |

**How does tree search visual choice improve slide quality?** To assess the contribution of the tree-search visual choice module, we conduct an ablation experiment, as shown in Table 5. In line with prior work on slide generation [38], we assess the generated slides using a VLM on a 1–5 scale across content, design, and coherence. The results show a pronounced decline in design quality

Table 5: **Evaluation result on slide quality**.

| Method | Content (↑) | Design (↑) | Coherence (↑) |
| --- | --- | --- | --- |
| HumanMade | **4.43** | **2.85** | 2.73 |
| PPTAgent$_{Qwen7B}$ [38] | 3.43 | 1.57 | 1.29 |
| PaperTalker$_{Qwen7B}$ | 4.00 | 2.53 | 3.11 |
| PPTAgent$_{GPT4.1}$ [38] | 4.07 | 2.02 | 2.06 |
| PaperTalker$_{GPT4.1}$(w/o Tree Search) | 4.33 | 2.73 | 3.84 |
| PaperTalker$_{GPT4.1}$ | 4.34 | **2.85** | 3.84 |

when layout refinement is removed, highlighting **the tree-search visual choice module as a key component for slide creation** (*i.e.,* resolving overfull issues), referring to Figures 8 for visualization.

## 6 CONCLUSIONS

This work tackles the long-standing bottleneck of presentation video generation by agent automation. With Paper2Video, we provide the first comprehensive benchmark and well-designed metrics to rigorously evaluate presentation videos in terms of quality, knowledge coverage, and academic memorability. Our proposed PaperTalker framework demonstrates that automated generation of ready-to-use academic presentation videos is both feasible and effective, producing outputs that closely approximate author-recorded presentations while significantly reducing production time by 6 times. We hope our work advances AI for Research and supports scalable scholarly communication.

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

# Appendix

**Contents**

# A    Checklist

## A.1   The Use of Large Language Models

In our work, LLMs are used for following aspects:

- Using an LLM to help with paper writing. We use GPT5 to help optimize language, correct grammar and write LaTeX table code.

- Using an LLM as a research assistant. We use GPT5 to help search related works.

- Using an LLM in our methods and experiment. This is described in the paper.

## A.2   Ethics Statement

We confirm that our study did not use any sensitive data where all data are public available. We have conducted this research and reported our findings responsibly. All results are presented transparently, including both performance gains and any observed limitations. We have diligently cited all relevant prior work and data sources to give proper credit and context. By following best practices in documentation and research integrity, we aim to contribute positively to the scientific community while upholding the highest ethical standards.

## A.3   Reproducibility statement

We are committed to ensuring the reproducibility of our results. All code and data needed to reproduce the experiments will be made publicly available. We will release this repository openly with an appropriate open-source license upon publication. The datasets used in our experiments are standard public benchmarks for language modeling and understanding (e.g., widely-used corpora and evaluation sets). These resources are readily accessible to other researchers.

# B    Evaluation Metrics

## B.1   IP Memory

We propose a novel metric to evaluate how well an audience retains a work after watching its presentation video. Motivated by **real-world conference interactions**, this metric assesses whether an audience member, after viewing several presentation videos, can recall the work and pose a relevant question when meeting the author.

To operationalize this, we construct video–question pairs by sampling a five-second clip from each presentation video and selecting a corresponding understanding-level question from PresentQuiz. A VideoLLM serves as the audience proxy: it is presented with four randomly sampled video–question pairs, where the videos and questions are shuffled, together with an image of one speaker as the query. The model is then asked to identify the relevant question to pose, and the accuracy quantifies the IP Memory score. Higher recall accuracy indicates that the generated results are more impressive and hold greater potential for lasting impact.

# C    Experiment

## C.1   Video Results

Video results please refer to the supplementary materials.

## C.2   Results of Tree Search Visual Choice

Figure 8 illustrates the slides before and after applying tree search visual choice refinement. The refinement **resolves the overfull issues** and substantially improves slide quality, indicating that this module **plays a crucial role in layout adjustment**.

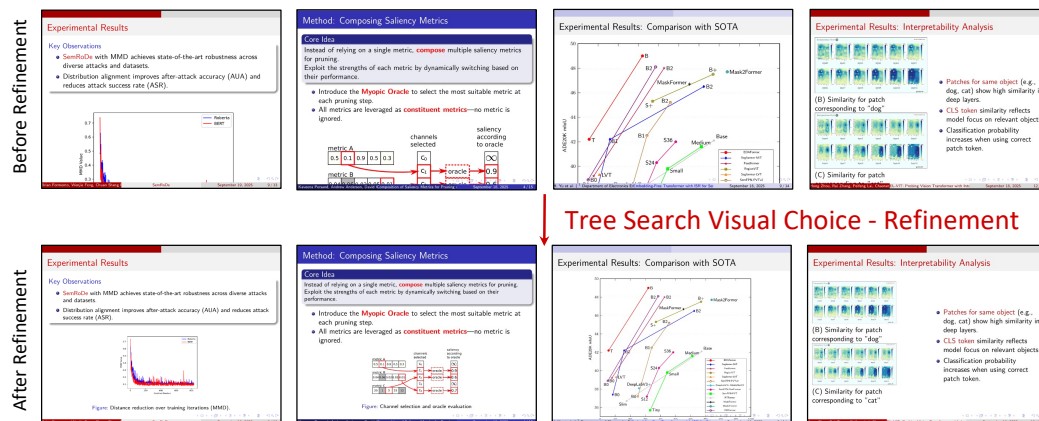

Figure 8: **Slide Visualization of Tree Search Visual Choice.** The first row shows slide results before layout refinement, while the second row shows their corresponding slides after refinement.

# D  Prompts

---

● **Prompt:** Slide Generation

**System Prompt:** Please generate a complete English PPT introduction based on the following TeX source text content, using LaTeX Beamer. The specific requirements are as follows.

**Content structure:**
- The PPT should contain the following chapters (arranged in order), and each chapter must have a clear title and content:
- Motivation (research background and problem statement)
- Related work (current status and challenges in the field)
- Method (core technical framework) [The content of the method needs to be introduced in detail, and each part of the method should be introduced on a separate page]
- Innovation (differentiation from existing work)
- Experimental method (experimental design and process)
- Experimental setting (dataset, parameters, environment, etc.)
- Experimental results (main experimental results and comparative analysis)
- Ablation experiment (validation of the role of key modules)
- Deficiencies (limitations of current methods)
- Future research (improvement direction or potential application)
- End slide (Thank you)

**Format requirements:**
- Use Beamer's theme suitable for academic presentations, with simple color matching.
- The content of each page should be concise, avoid long paragraphs, and use itemize or block environment to present points. The title page contains the paper title, author, institution, and date.
- Key terms or mathematical symbols are highlighted with alert{}.

**Image and table processing:**
- All image paths are given, and relative paths are used when citing, the picture names must "be consistent with the name in tex file".
- Images should automatically adapt to width, and add titles and labels
- Experimental result tables should be extracted from the source text, formatted using tabular or booktabs environments, and marked with reference sources ( "as shown in table").

---

**Code generation requirements:**

- The generated LaTeX code must be complete and can be compiled directly (including necessary structures).

- Mark the source text location corresponding to each section in the code comments (for example,

- If there are mathematical formulas in the source text, they must be retained and correctly converted to LaTeX syntax (such as $y = f(x)$).

**Other instructions:**

- Image content should be read from the tex file, and the source name should be used directly without arbitrary modification. Image references should use real image names and should not be forged;

- Table content should first extract real data from the source document.

- All content should be in English.

- If the source text is long, it is allowed to summarize the content, but the core methods, experimental data and conclusions must be retained.

- To enhance readability, a transition page can be added (for example, "This section will introduce the experimental part").

- Perfer more images than heavy text. **The number of slides should be around 10.**

- **& in title is not allowed which will cause error "Misplaced alignment tab character &** **Pay attention to this "error: !File ended while scanning use of frame**

- Only output latex code which should be ready to compile using tectonic(simple verson of TeX Live). Before output check if the code is grammatically correct.

---

● **Prompt:** Error Correction

**System Prompt:** You are given a LaTeX Beamer code for the slides of a research paper and its error information. Correct these errors *without changing* the slide content (text, figures, layout).

**Instructions:**

- Apply the minimal edits required to make the file compile: add missing packages, close/open environments, balance braces, escape special characters, fix math delimiters, resolve duplicate labels, and correct obvious path or option typos.

- Do *not* paraphrase or delete text; do *not* change figure/table content, captions, labels, or layout semantics.

- Keep all image/table file names and relative paths as given; do not invent or rename assets.

- Preserve the original Beamer theme, colors, and structure.

- Ensure the final output compiles with **Tectonic**; close all environments and avoid undefined commands.

**Output (strict):** Output *only* the corrected LaTeX source, beginning with `beamer` and ending with `document`; no extra commentary.

**Prompt:** MSTS Judge

**System Prompt:** You are a slide layout judge. You see four slides A–D in a 2×2 grid: A (top-left), B (top-right), C (bottom-left), D (bottom-right).

**Definitions**

- **Overfull:** any part of the figure or its caption is clipped, outside the frame, or overlapped/hidden.

- **Coverage:** among non-overfull options, larger visible content with less empty background is better.

- **Risk:** risk of overfull decreases from A → D (A largest, D smallest).

- **Coverage trend:** coverage decreases from A → D.

**Rules (judge only the given images)**

1. Disqualify any option with overfull (caption must be fully visible).

2. From the remaining, pick the one with the greatest coverage.

3. Practical method: scan **A → B → C → D**; choose the *first* slide in that order that is not overfull.

**Output only (strict; do *not* output `` ```json ``):**

```
{
"reason": "concise comparison",
"choice": "A" | "B" | "C" | "D"
}
```

**Prompt:** Slide Script with Cursor Positions

**System Prompt:** You are an academic researcher presenting your own work at a research conference. You are provided with a sequence of adjacent slides.

**Instructions:**

- For each slide, write a smooth, engaging, and coherent first-person presentation script.

- Clearly explain the *current* slide with academic clarity, brevity, and completeness; use a professional, formal tone and avoid content unrelated to the paper.

- Each sentence must include *exactly one* cursor position description drawn from the *current slide* and listed in order, using the format `script | cursor description`. If no cursor is needed for a sentence, write `no`.

- Limit the total script for each slide to **50 words** or fewer.

- Separate slides using the delimiter `###`.

**Output Format (strict):**

```
sentence 1 | cursor description
sentence 2 | cursor description
...
###
sentence 1 | cursor description
...
```

**Prompt:** Meta Similarity

**System Prompt:** You are an evaluator. You will be given two presentation videos of the same talk: (1) a human-presented version and (2) an AI-generated version. Evaluate *only* the slides and subtitles; ignore the presenter's face, voice quality, background music, camera motion, and any non-slide visuals.

**Inputs You May Receive**
- Human video (and optionally its slide images and subtitles/transcript)
- AI video (and optionally its slide images and subtitles/transcript)

**Evaluation Scope (focus strictly on slides + subtitles)**

1. **Slide Content Matching:** Do AI slides convey the same key points and comparable layout/visual elements (titles, bullets, diagrams, tables, axes annotations) as the human version?

2. **Slide Sequence Alignment:** Is slide order consistent? Any sections missing, added, or rearranged?

3. **Subtitle Wording Similarity:** Do AI subtitles reflect similar phrasing/terminology and information as the human speech/subtitles? Focus on semantic equivalence; minor style/spelling differences do not matter.

4. **Slide–Subtitle Synchronization:** Within the AI video, does narration/subtitle content match the on-screen slide at the same time? Does this broadly align with the human presenter's per-slide content?

**Evidence-Only Rules**
- Base the judgment solely on the provided materials (videos, slides, subtitles). Do *not* use outside knowledge.
- If some inputs are missing (*e.g.*, no subtitles), judge from what is available and briefly note the missing piece in the Reasons.

**Relaxed Scoring Rubric (0–5)**
- **5** — Nearly identical: slides and subtitles closely match the human version in content, layout, sequence, and timing; wording is near-paraphrase.
- **4** — Highly similar: only minor layout/phrasing differences; content, order, and alignment clearly match.
- **3** — Moderate differences yet same core content: several layout/wording/sequence deviations but main sections and key points are preserved. (Leniency: borderline cases between 2 and 3 *round up* to 3.)
- **2** — Partial overlap: substantial omissions/rearrangements or subtitle drift; multiple slide mismatches or sync issues.
- **1** — Minimal overlap: only a few matching fragments; most slides/subtitles diverge.
- **0** — No meaningful match: AI slides/subtitles do not correspond to the human version.

*Lenient mapping: if borderline between adjacent levels, choose the higher score. If computing subscores, average and **round up** to the nearest integer in [0,5].*

**Output Format (STRICT; exactly one line)**

```
Content Similarity: X/5; Reasons
```

Where `X` is an integer 0–5 from the rubric, and `Reasons` is 1–3 short sentences referencing content, sequence, wording, and synchronization as relevant.

**● Prompt:** PresentArena

**System Prompt:** You are an expert in evaluating academic presentation videos. You are given two videos (Video A and Video B) on the same research topic. Evaluate each video independently and then decide which is better, or if they are basically the same (preferred when not confident).

**Evaluation Criteria**
- **Content Clarity**: Are key ideas and findings clearly explained?
- **Speaker Delivery**: Is the speaker confident, fluent, and engaging?
- **Visual Aids**: Are slides/visuals clear, helpful, and well-integrated?
- **Structure & Pacing**: Is the talk logically organized and appropriately paced?
- **Audience Engagement**: Does the speaker maintain interest and attention?

**Steps**
1. **Step 1:** Write a short (1–2 sentence) evaluation of **Video A** based on the criteria.
2. **Step 2:** Write a short (1–2 sentence) evaluation of **Video B** based on the criteria.
3. **Step 3:** Decide which video is better, or if they are basically the same (prefer "Same" if not confident).

**Output Format (Strict; only these three blocks):**

```
Step 1:
[1-2 sentences evaluating Video A]

Step 2:
[1-2 sentences evaluating Video B]

Step 3:
Final Judgment:
[A] | [B] | [Same]

Reason: [One concise sentence justifying the judgment based on Steps1-2.]
```

**● Prompt:** PresentationQuiz

**System Prompt:** You are an answering agent. You will be provided with: 1) a presentation video of a paper, and 2) a JSON object called `"questions"` containing multiple questions, each with four options (A–D). Analyze the video thoroughly and answer each question *solely* based on the video content (no external knowledge). Do not reference timesteps that exceed the video length.

**Instructions:**
- For each question, if the video provides sufficient evidence for a specific option (A, B, C, or D), choose that option.
- Include a brief reference to where in the video the evidence appears (*e.g.*, "Top-left text", "Event date section").
- Rely only on the video; do not use outside context.
- Provide an answer entry for *all* questions present in `"questions"`.

**Template (steps to follow):**
1. Study the presentation video together with `"questions"`.
2. For each question, determine whether the video clearly supports one of the four options; if so, pick that answer.

3. Provide a brief reference indicating where in the video you found the evidence.

4. Format the final output strictly as a JSON object with the following pattern (and no extra keys or explanations).

**Output Format (strict):**

```
{
  "Question 1": { "answer": "X", "reference": "some reference" },
  "Question 2": { "answer": "X", "reference": "some reference" },
  ...
}
```

**questions payload:**

```
{{questions}}
```

