# OpenReview forum: "Paper2Video: Automatic Video Generation from Scientific Papers"
_ICLR.cc/2026/Conference — Submitted to ICLR 2026_

### Official Review · Reviewer_DjH8 · 2025-10-21

**Soundness:** 2
**Presentation:** 3
**Contribution:** 2
**Rating:** 4
**Confidence:** 4

**Summary:**

This paper proposes a benchmark called Paper2Video and a multi-agent framework, PaperTalker, for automatically generating academic presentation videos from academic papers. The authors first constructed a benchmark dataset consisting of 101 papers paired with author-recorded videos and proposed four new evaluation metrics (Meta Similarity, PresentArena, PresentQuiz, and IP Memory) to measure the quality of automatically generated videos. The authors then designed the PaperTalker framework, improving efficiency through a modular parallel generation strategy. Experimental results show that this approach outperforms existing methods (such as PresentAgent and Veo3) across multiple metrics and approaches the performance of manually produced videos in human evaluations.

**Strengths:**

1. The paper defines academic presentation video generation as an AI for Research task and provides a benchmark and evaluation criteria.
2. Four custom metrics (Meta Similarity, PresentArena, PresentQuiz, and IP Memory) cover different dimensions of academic presentation videos, from content fidelity, audience comprehension, and memorability.
3. The proposed multi-agent PaperTalker framework is well-designed, encompassing slide layout optimization, speech generation, talking head synthesis, and cursor grounding.

**Weaknesses:**

1. The paper claims that videos generated by PaperTalker are "more faithful and informative than existing baselines." However, the user study and subjective evaluations are based on a small sample size (only 10 participants), and the evaluator backgrounds, scoring criteria, and statistical significance are not disclosed, which undermines the credibility of the conclusions.
2. Four custom metrics (especially PresentArena and PresentQuiz) rely heavily on VideoLLM's judgments, but the paper does not detail the bias control mechanisms. For example, PresentArena's pairwise comparisons may be affected by LLM bias, and no consistency or confidence intervals are reported.
3. Paper2Video includes only 101 sample papers, primarily from AI conferences (such as NeurIPS, ICLR, and CVPR), with limited field coverage, which limits the model's generalizability. There is a lack of cross-disciplinary validation samples (such as biology and physics).
4. In comparisons with baselines such as PresentAgent and Veo3, the authors did not explicitly control the consistency of input conditions (such as paper length, slide templates, and audio quality), which may affect the fairness of the conclusions. In particular, Veo3 is a natural video generator, and its task objectives are different.
5. While Figure 7 and the related discussion demonstrate visual results, they lack in-depth qualitative analysis, such as a discussion of the interpretability of slide layout optimization, a classification of generated error types, or an analysis of specific failure cases.
6. The paper assumes that the cursor remains stationary within each sentence and only moves between sentences. This is significantly inconsistent with real human speech. Speakers often dynamically move the cursor throughout a sentence to emphasize specific words, parts of a formula, or areas of a diagram. This simplified, discrete movement may feel unnatural to the audience. While the paper experimentally demonstrates that the presence of the cursor helps locate information, it does not evaluate the quality of the generated cursor trajectory itself, such as whether it compares to human-generated trajectories in terms of naturalness and effectiveness.

**Questions:**

1. In the PresentArena metric, VideoLLM is used as an automated judge. Please explain how you control the consistency of LLM responses? Do you control for prompt or sequence bias?
2. Are the full LaTeX source code, slides, and videos for each example publicly available in the Paper2Video dataset? Are copyright and ethical approvals obtained?
3. What is the VLM scoring metric in the Tree Search Visual Choice module? Will the computational complexity of this module become a bottleneck in larger-scale generation?
4. Please provide more details about PresentQuiz question generation: number of questions, coverage, difficulty control, and whether it was manually validated?
5. Does the human evaluation use statistical tests (e.g., t-test or Wilcoxon) to verify the significance of the differences between PaperTalker and human-generated videos?
6. Regarding the speaker videos, the paper mentions the use of models such as Hallo2 and FantasyTalking. These models primarily generate head and upper body movements. Does your system consider body language, such as gestures? Although the paper mentions that FantasyTalking supports upper body movements, it does not demonstrate or discuss the relevant generation effects.

---

> ### Author Response · Authors · 2025-11-21
> **Response (Part 1)**
>
> Thank you for recognizing the value of our work and for providing constructive and thorough feedback! We value your insights and hope our response addresses your comments.
>
>
> **W1**: *Human Study is Limited*
>
> Thank you for your suggestion. We **have added 15 additional participants**, resulting in a total of **25**. Since the primary audience of academic presentation videos is the research community, all participants are **graduate students with diverse backgrounds** (e.g., engineering, social sciences) to ensure the reliability. The instructions are shown below:
> > You will be given five academic paper presentation videos and asked to provide an overall ranking for these videos.
> When ranking, please consider the following four aspects: **Accuracy** (How complete, correct and reliable the content is considering that it represents a paper); **Fluency** (How coherent and smooth the narration is); **Visual Quality** (How clear and visually appealing the video is); **Audio Naturalness** (How natural and pleasant the audio sounds). Your ranking should reflect your overall impression across these four dimensions. To ensure objective evaluation:
> 1 Please watch all four videos carefully before ranking.
> 2 Base your decision solely on what you see and hear in the videos.
> 3 Evaluate each video independently before comparing them.
> 4 Avoid personal biases unrelated to the video content.
> Your responses are anonymous and will be used only for academic research.
>
> The results are shown below:
>
> | Human | PaperTalker | PresentAgent | VEO3 | Wan2.2 |
> |--------------------|----------------------------|-----------------------------|---------------------|------------------------|
> | 4.32 ± 0.79          | 3.76 ± 1.21               | 3.08 ± 0.56                 | 2.84 ± 1.12         | 1.00 ± 0.00            |
>
> The paired t-test between Human and PaperTalker shows no statistically significant difference **(t = 1.62, p = 0.12)**. Although Human receives slightly higher mean scores, the p-value indicates that this gap can be attributed to random variability, suggesting that **PaperTalker’s performance is statistically comparable to Human annotators**. In contrast, the paired t-test between PaperTalker and PresentAgent reveals a significant difference **(t = 2.53, p = 0.018)**. PaperTalker consistently achieves higher scores, **indicating a reliable performance advantage over PresentAgent across the evaluated samples**.
>
> **W2**: *Bias Control in Evaluation*
>
> We have considered the potential evaluation bias brought by VideoLLM.
> 1. First, for all the methods we use **the same prompt during evaluation** which is shown in the **Section C of Appendix**.
> 2. Especially for PresentArena, we consider the sequential bias in pairwise comparisons. As stated in Sec 3.3, for each pair, the model is queried twice **in opposite orders: (A, B) and (B, A)**. This procedure reduces bias brought by input position.

---

> ### Author Response · Authors · 2025-11-21
> **Response (Part 2)**
>
> **W3**: *Benchmark Coverage*
>
> ### (i) We begin with AI conferences mainly for two reasons:
> 1. **Our datatset already offer representativeness and diversity.**
> Our dataset spans major AI subfields—including CV, NLP, and general ML—and covers papers of varying lengths with rich structures (text, figures, tables, and mathematical content). These structures can also be observed from other fields such as such as biology and physics. Besides, these papers also enforce rigorous peer review, ensuring that the source papers are clear, well-organized, and of consistently high quality, making them a practical foundation for studying paper-to-video generation.
>
> 2. **AI conferences uniquely provide high-quality author-recorded presentations.**
> Curating aligned paper–video pairs is non-trivial. Outside of AI, many fields (e.g., Physics, Sociology) rarely publish a dedicated presentation video per paper, which limits the feasibility of large-scale collection. Even within AI conferences, we still face challenges such as incomplete metadata and inconsistent auxiliary materials; therefore, we manually filter papers with relatively complete information and supplement missing fields to maintain data reliability.
>
> ### (ii) Why Paper2Video contains 101 samples?
> In terms of dataset size and robustness, we want to mention that Paper2Video is a **long-horizon agentic task**, introducing substantial challenges and distinctions compared to prior VLM benchmarks. While our benchmark contains 101 samples, each sample in Paper2Video is much heavier than instances in standard VLM task benchmarks. Below is a comparison of Paper2Video (101 samples) with a popular VLM task MM-Vet [1] (218 samples):
>
> |                                 | **VLM task (MM-Vet)**            | **Agentic task (Paper2Video)**     |
> |---------------------------------|----------------------------------|--------------------------------------|
> | **Input**                       | Image + Question                 | Paper + Image + Audio                                |
> | **Input context length (avg.)** | 1 image + <1k tokens             |  > 22 images > 12k tokens            |
> | **Output length (avg.)**        | <100 tokens                      |  > 4mins video > 5,000 image > 4.6 million tokens         |
> | **Per sample time**             | Within seconds (single VLM gen.) | > 40 mins             |
> | **Number of Samples**           | 218                              | 101                                  |
>
> Given these increases in context, output, and runtime, scaling Paper2Video is far more resource-intensive than scaling existing VLM benchmarks. Our 101 carefully constructed samples already provide **broad coverage while keeping evaluation feasible and reproducible**.
>
> By contrast, MM-Vet can be completed within minutes (~a few seconds per item × 218 items), whereas a full Paper2Video run requires more than 40 hours per model (>40 min per instance × 101 instances). Our current scale, therefore, strikes a pragmatic balance: large enough to expose systematic trends, yet light enough to encourage broad adoption.
>
> However, we greatly appreciate your viewpoint, and we will continue to expand the benchmark and release additional samples as resources permit in future work.
>
> **W4**: *Input Conditions in Baseline Comparison*
>
> ### (i) We have explicit control the different settings
> As the baselines are developed under different settings, we controlled input conditions as consistently as possible:
>
> **1. Paper length.** Paper length is kept identical for PresentAgent and PaperTalker. Since Veo3 does not support PDF or long-text input, we first use an LLM to generate a summary of the paper (≈3k words) and use it as its generation prompt.
>
> **2. Slide templates.** PaperTalker does not require a slide template. For PresentAgent, we use the default template provided in its official codebase to avoid introducing any bias from external template design.
>
> **3. Audio quality.** PresentAgent and Veo3 do not support voice customization; therefore, we do not provide any audio samples as input for these baselines. Moreover, the evaluations of PresentQuiz and PresentArena do not rely on voice customization.
>
> Overall, these controls ensure that all models operate under comparable and fair conditions, isolating differences in performance to model capabilities rather than input inconsistencies. We follow a best-effort reproducibility principle throughout our comparisons.
>
> ### (ii) Is Veo3 for presentation generation proper?
>
> Although Veo3 is primarily designed for natural-video generation rather than slide-based presentations, we included it as an exploratory comparison, motivated by the interesting trend in which general video or image generators (e.g., **NanoBanana for poster creation**) are applied beyond their natural video or image generation. While Veo3 is not directly aligned with our task, its performance provides additional perspective on the inherent difficulty of presentation generation.

---

> ### Author Response · Authors · 2025-11-21
> **Response (Part 3)**
>
> **W5**: *In-depth Analysis about Figure 7*
>
> Thank you for your suggestion. Here we provide an in-depth analysis of Figure 7.
> - **Veo3**. The video generated by Veo3 exhibits high visual quality but has a very short duration and contains many incorrect words. These errors mainly fall into two categories: **blurred or indistinct characters**, and **typographical mistakes** (e.g., “Labuel,” “sofft”). This observation highlights the shortcomings of current natural video generation models when producing text-heavy content, which further motivates us to address academic presentation video generation using a multi-agent framework that renders slides through Beamer code.
> - **PresentAgent**. While PresentAgent produces human-like presentation videos, the generated slides may lose important information from the paper and exhibit many layout issues (e.g., overfull tables or images, and dense text without bullet points). These problems stem from the fact that **the VLM is insensitive to subtle numerical changes** during iterative slide parameter adjustment, leading to unstable or suboptimal layout refinement.
> - **PaperTalker**. PaperTalker generates the most human-like presentation videos with well-structured slides, talker, and cursor highlight. The fine-grained slide layout is ensured with the following reasons:
>     1. LaTeX’s compiler **automatically optimizes the placement** of text blocks and figures, so the only parameters that need to be specified are the layout style (e.g., single-column or double-column) and the element sizes.
>     2. Tree-Search Visual Choice ensures reasonable element-size selection. Specifically, we first generate neighboring candidate slide parameters using rule-based heuristics, and then use a VLM to score these candidates and select the best one. This design is motivated by the observation that VLMs are insensitive to subtle numerical changes, and therefore **decouples parameter adjustment from optimal candidate selection**, leading to more stable and reliable layout refinement. More visualizations are shown in Figure 8.
>
> We will further include these analyses in our future version.
>
>
> **W6**: *Cursor Generation*
>
> We acknowledge that generating high-quality cursor trajectories is a challenging problem. To the best of our knowledge, our work is **the first** to introduce **cursor generation for academic presentation videos**. Even generating sentence-level cursor trajectories in our setting is highly non-trivial, as it requires precise **spatial grounding with slide content** and **temporal grounding with the speech narrative**.
>
> ### (i) The rationality of sentence-level cursor anchoring
> We agree that real presenters may move the cursor within a sentence to highlight specific words, formulas, or regions of a diagram. At the same time, human presenters also **exhibit notable pauses and stable points**, which makes sentence-level cursor anchoring a reasonable first step.
>
> ### (ii) Directly comparing to human's is difficult to control.
> Our experiments focus on evaluating the **functional role** of the cursor: whether it helps viewers more effectively locate relevant information on the slides. This aligns with the primary purpose of the cursor in our system. In real presentations, cursor movements do not follow a single canonical pattern; they **depend heavily on slide design, presenter habits, and the flow of the narration**. As a result, directly comparing our generated trajectories to “human” trajectories is difficult to control (e.g., **requiring identical slides and transcripts**) and may introduce **subjective biases** driven by individual behavioral differences rather than reflecting meaningful quality differences. For these reasons, we adopt a task-centric evaluation: assess whether the generated cursor improves information localization, rather than enforcing similarity to any specific human motion pattern.
>
> In future work, we plan to explore more flexible cursor trajectories (e.g., allowing the agent to decide when to move the cursor), with the goal of achieving more realistic and human-like presentation behavior.
>
> **Q1**: *PresentArena Evaluation Control*
>
> Thanks for pointing this out. Maintaining consistent prompts is crucial for ensuring both reproducibility and fair evaluation across methods.
>
> For PresentArena, all the methods we use the **same prompt** during evaluation. Moreover, we consider the **sequential bias** in pairwise comparisons. As stated in Sec 3.3, for each pair, the model is **queried twice in opposite orders**: (A, B) and (B, A). This procedure reduces bias brought by input position. We also control the consistency LLM output in a certain format in the prompt. Details please refer to Section C of the Appendix.

---

> ### Author Response · Authors · 2025-11-21
> **Response (Part 4)**
>
> **Q2**: *Copyright*
>
> All materials in Paper2Video are accessed through their original, publicly available sources. For each example, we provide (i) the **ArXiv link** to the full LaTeX project and paper PDF, and (ii) the **URLs to the author-uploaded presentation videos and slides** hosted on platforms such as SlidesLive and YouTube. We **do not redistribute** any copyrighted content; all materials are viewed and downloaded directly from the original hosting platforms.
>
> **Q3**: *Scoring Metric and Computational Complexity Tree Search Visual Choice*
>
> As described in Section 4.1, Tree Search Visual Choice first explores a set of nearby layout candidates (e.g., variations in text size or figure size). Each candidate is then rendered into a slide, and all rendered slides are concatenated into a single image that is passed to the VLM. The prompt is provided in Appendix D (MSTS Judge). Since we perform only **a single inference per slide with layout issues**(e.g., overfull), the computational cost grows linearly with the number of slides, and therefore **does not become a bottleneck** even at larger generation scales.
>
> We further compare our Tree Search Visual Choice with the slide refinement strategy used in PresentAgent. It performs multiple refinement steps per slide, resulting in higher computational cost, especially as the number of slides grows. In contrast, our method executes a single VLM inference for each slide with layout issues, yielding a more scalable procedure.
> | Method                           | Complexity per Presentation | Reasoning Steps per Slide                            | Overall Scaling                         |
> |----------------------------------|-----------------------------|--------------------------------------------------------|-------------------------------------------|
> | **PresentAgent**                 | O(nk)                   | Requires \(k\) iterative refinement or decision steps | Grows linearly in \(n\) and proportionally in \(k\) |
> | **Tree Search Visual Choice (Ours)** | O(n)                    | One inference for slides with layout issues           | Linear in \(n\), independent of \(k\)    |
>
> This efficiency difference makes our method more suitable for long presentations or large-scale generation settings.
>
> **Q5**: *Human Study*
> > Does the human evaluation use statistical tests (e.g., t-test or Wilcoxon) to verify the significance of the differences between PaperTalker and human-generated videos?
>
> Please refer to **W1**.
>
> **Q6**: *Presenter Gesture*
>
> Thank you for the question. Body language and gesture modeling is indeed an interesting direction that could make generated presentations more expressive and speaker-like. Our current method focuses on **a practical setting aligned with real academic presentations**, which typically include only the head region. This design choice also offers practical benefits: limiting generation to the head region tends to provide more stable long-duration video synthesis, while keeping the computational requirements manageable for large-scale use.
>
> Looking forward, developing TED-level full-body, multi-shot presentation generation remains a challenging and promising direction, and we see substantial potential for future work in this area.

---

> ### Author Response · Authors · 2025-11-21
> **Response (Part 5)**
>
> **Q4**: *Details about PresentQuiz*
>
> In PresentQuiz, we generate **100 questions** for each paper with **50 each about details and understanding**. We control coverage and difficulty during the generation of the questions as follows:
> - For detailed questions, we use the following prompt for coverage and difficulty control:
> > Write 50 factual, answerable questions.
>      • Each question must map to one clear sentence/phrase in the paper text.
>      • No duplicate or near-duplicate wording.
>      • **Vary difficulty from easy “headline” facts to specific numeric or procedural details.**
>      • **Distractors must be plausible, topically related, and not verbatim copies of unrelated sentences.**
> Distribute the 50 questions across the following aspects. Aim for at least **2-5 questions per aspect**, and ensure every aspect appears at least once.
>
>        A. Title & authorship (title, author names, affiliations, keywords)
>        B. Motivation / problem statement / research gap
>        C. Objectives or hypotheses
>        D. Dataset(s) or experimental materials
>        E. Methodology (algorithms, model architecture, workflow steps)
>        F. Key parameters or hyper-parameters (values, settings)
>        G. Evaluation metrics or criteria
>        H. Quantitative results (numbers in tables, charts)
>        I. Qualitative findings, figures, or illustrative examples
>        J. Comparative or ablation study results
>        K. Conclusions, implications, or contributions
>        L. Limitations or future work
>        M. Definitions of domain-specific terms or abbreviations
>
> - For understanding questions, we use the following prompt for coverage and difficulty control:
> >   Draft 50 factual questions that probe the reader's global grasp of the
>      paper (e.g., “What problem does the study address?”).
>      • Avoid low-level numeric settings, code snippets, or reference lists.
>      • Vary wording and avoid duplicates.
>      • **Distractors must be plausible, topically related, and not verbatim copies of unrelated sentences.**
>     Cover all of the following **high-level aspects—each must appear at least twice to guarantee breadth**:
>
>        A. Research domain & background context
>        B. Central problem / motivation / research gap
>        C. Primary goal, hypothesis, or research question
>        D. Key contributions or novelty statements
>        E. Overall methodology or workflow (summarized)
>        F. Principal findings or headline quantitative results
>        G. Qualitative insights or illustrative examples
>        H. Implications, applications, or significance
>        I. Limitations or future-work directions
>        J. Main conclusions or take-home messages
>
> In total, we generate **10,100 questions** for 101 papers. To ensure balanced coverage and well-controlled difficulty, we statistically analyze the distributions for the **Detail** and **Understanding** question categories as shown below:
> | Detail Category | A     | B     | C     | D     | E     | F     | G     | H     | I     | J     | K     | L     | M      |
> |-----------------|-------|-------|-------|-------|-------|-------|-------|-------|-------|-------|-------|-------|--------|
> | Freq (%)        | 8.74% | 7.04% | 5.84% | 7.36% | 8.38% | 7.50% | 6.68% | 9.22% | 7.00% | 6.30% | 8.16% | 6.82% | 10.96% |
>
> | Understanding Category | A     | B     | C     | D     | E     | F     | G     | H     | I     | J     |
> |------------------------|-------|-------|-------|-------|-------|-------|-------|-------|-------|-------|
> | Freq (%)               | 9.68% | 9.40% | 9.82% | 9.96% | 10.56%| 10.56%| 10.02%| 9.92% | 10.28%| 9.80% |
>
> Overall, the distributions are well balanced across categories, ensuring both **broad topic coverage** and **controlled difficulty levels** in the constructed question set. We do not perform exhaustive manual validation for all 10,100 questions, as this would require a substantial amount of human labor. However, we carefully design the generation prompts, include content-grounding constraints, and empirically verify samples during development. Combined with the question statistical analysis, these measures help ensure the overall quality, coverage, difficulty, and consistency of the PresentQuiz questions.

---

> ### Author Response · Authors · 2025-11-25
>
> Dear Reviewer DjH8,
>
> Thank you once again for your feedback!
>
> We would greatly appreciate it if you could review our response to ensure it adequately addresses your concerns. We remain fully dedicated to clarifying any remaining points and would welcome any further discussion to ensure all your questions are thoroughly answered.
>
> Thank you for your time and consideration.
>
> Best,
>
> Authors of Paper2Video

---

> > ### Comment · Reviewer_DjH8 · 2025-11-25
> >
> > Thank you for the responses. I appreciate the additional analyses and clarifications provided. I have reviewed your updates carefully. Given the new information, I will keep my score, mainly because I believe the contribution is interesting but still limited by the current dataset scale and evaluation design.
> >
> > Thank you again for the effort in addressing the comments.

---

> ### Author Response · Authors · 2025-11-28
>
> >Summary: the contribution is interesting but still limited by the current dataset scale and evaluation design.
>
> Thank you for the thoughtful follow-up and for carefully reviewing our updates. We would like to offer two key clarifications regarding the data scale and evaluation design.
>
> 1.  **Scale of Long-horizon Agentic tasks.** Paper2Video is a long-horizon agentic task, where each instance contains **>22 images, >12k tokens of paper content, and >4 minutes of synchronized video (~4.6M tokens)**. A single model run already requires **over 40 minutes**, making large-scale expansion substantially more resource-intensive than conventional VLM benchmarks. Under this cost profile, **101 samples represent a large effective evaluation load**, and our scale is aligned with prior multimodal-generation agentic benchmarks such as Paper2Poster[1] (**100** samples), Paper2Code[2] (**90** samples) and PaperBench[3] (**20** samples).
>
> 2. **Define a good presentation is non-trivial**. To the best of our knowledge, we are the **first** to explicitly articulate **what defines a “good” academic presentation video** and to build evaluation metrics grounded in how human viewers actually process such videos. Specifically, we assess **(i)** slide and speech quality, **(ii)** information delivery effectiveness, and **(iii)** audience academic memory. These complementary perspectives provide a holistic measure of how well a generated presentation functions both visually and communicatively. Furthermore, we expanded **a larger-scale human study**, providing additional evidence that our evaluation captures meaningful differences across models and **aligns with real viewer experience**.
>
>
> ---
> We hope this clarifies both the adequacy of our dataset scale and the rigor of our evaluation design. We remain fully dedicated to clarifying any remaining points and would welcome any further discussion to ensure all your questions are thoroughly answered. If you feel that our rebuttal and revisions have strengthened the contribution of the paper, we would be truly grateful if you might consider adjusting your score to champion its acceptance.
>
> [1] Pang, Wei, et al. “Paper2Poster: Towards Multimodal Poster Automation from Scientific Papers.” Advances in Neural Information Processing Systems (2025).
>
> [2] Seo, Minju, et al. "Paper2code: Automating code generation from scientific papers in machine learning." arXiv preprint arXiv:2504.17192 (2025).
>
> [3] Starace, Giulio, et al. "PaperBench: Evaluating AI’s Ability to Replicate AI Research." Forty-second International Conference on Machine Learning.

---

### Official Review · Reviewer_cGYF · 2025-10-30

**Soundness:** 2
**Presentation:** 3
**Contribution:** 2
**Rating:** 4
**Confidence:** 4

**Summary:**

This work presents the Paper2Video benchmark dataset and the PaperTalker multi-agent system to convert papers into slide-based presentation videos. It uses four types of metrics to measure consistency with human works, preference, knowledge coverage, and memorability. The project demo shows the framework works.

**Strengths:**

- Directly turns papers into publishable presentation videos.
- Uses PaperTalker to generate academic-style slides, subtitles, voice-over, and a talking-head presenter, together with cursor-synchronized guidance, forming a closed loop.
- Uses personalized TTS and presenter-avatar video, and adds narration subtitles and cursor pointing, which can aid memory and visibility.

**Weaknesses:**

- Insufficient data scale: the benchmark is small (101), limiting coverage and statistical power.
- Comparative evaluation is incomplete: recent methods (e.g., EvoPresent, PreGenie, Preacher) are not included, making it hard to position SOTA. In addition, only 10 participants per paper rate the videos, so persuasiveness is limited.
- Limited style flexibility: using Beamer enforces norms but leaves limited room for personalization.

**Questions:**

- Increase the dataset size and include comparisons with the latest research projects.
- Add slide stylization features.
- In human evaluation, increase the number of participants per paper to improve the credibility of the scores.
- In Figure 7, the conclusion that human-made videos have no cursor may not generally hold.
- Two of the three automatic evaluations use (video) LLMs as the viewer/judge, which can couple the evaluation distribution with model preferences; consider adding larger-scale human evaluation or user-behavior metrics.

---

> ### Author Response · Authors · 2025-11-21
> **Response (Part 1)**
>
> Thank you for recognizing the value of our work and for providing constructive and thorough feedback! We value your insights and hope our response addresses your comments.
>
> **W1**: *Insufficient Data Scale*
>
> In terms of dataset size and robustness, we want to mention that Paper2Video is a **long-horizon** agentic task, introducing substantial challenges and distinctions compared to prior VLM benchmarks. While our benchmark contains 101 samples, each sample in Paper2Video is much heavier than instances in standard VLM task benchmarks. Below is a comparison of Paper2Video (101 samples) with a popular VLM task MM-Vet [1] (218 samples):
>
> |                                 | **VLM task (MM-Vet)**            | **Agentic task (Paper2Video)**     |
> |---------------------------------|----------------------------------|--------------------------------------|
> | **Input**                       | Image + Question                 | Paper + Image + Audio                                |
> | **Input context length (avg.)** | 1 image + <1k tokens             |  > 22 images > 12k tokens            |
> | **Output length (avg.)**        | <100 tokens                      |  > 4mins video > 5,000 image > 4.6 million tokens         |
> | **Per sample time**             | Within seconds (single VLM gen.) | > 40 mins             |
> | **Number of Samples**           | 218                              | 101                                  |
>
> Given these increases in context, output, and runtime, scaling Paper2Video is far more resource-intensive than scaling existing VLM benchmarks. Our 101 carefully constructed samples already provide **broad coverage while keeping evaluation feasible and reproducible**.
>
> By contrast, MM-Vet can be completed within **minutes** (~a few seconds per item × 218 items), whereas a full Paper2Video run requires **>40 hours** per model (40 min per instance × 101 instances). Therefore, our current scale strikes a pragmatic balance: large enough to expose systematic trends, yet light enough to encourage broad adoption.
>
> However, we greatly appreciate your viewpoint, and we will continue to expand the benchmark and release additional samples as resources permit in future work.
>
> [1] MM-Vet: Evaluating Large Multimodal Models for Integrated Capabilities. ICML 2024
>
> **W2**: *Baselines Lack Recent EvoPresent, PreGenie, Preacher*
> > Comparative evaluation is incomplete: recent methods (e.g., EvoPresent, PreGenie, Preacher) are not included, making it hard to position SOTA.
>
> Thank you for the information. The reference and publication time of the given three papers are shown below:
>
> Publish time: 21 Oct., 2025
> *[2] Liu, Chengzhi, et al. "Presenting a Paper is an Art: Self-Improvement Aesthetic Agents for Academic Presentations." arXiv preprint arXiv:2510.05571 (2025).*
>
> Publish time: 31 Aug., 2025
> *[3] Xu, Xiaojie, et al. "PreGenie: An Agentic Framework for High-quality Visual Presentation Generation." arXiv preprint arXiv:2505.21660 (2025).*
>
> Publish time: 14 Aug., 2025
> *[4] Liu, Jingwei, et al. "Preacher: Paper-to-video agentic system." Proceedings of the IEEE/CVF International Conference on Computer Vision. 2025.*
>
> All these papers are published after 1 Aug., 2025, which is within two monthes of the ICLR2026 submission deadline. However according to the ICLR2026 submission policy (https://iclr.cc/Conferences/2026/AreaChairGuide):
> >Q: Are authors expected to cite and discuss very recent work?
> >A: We consider papers contemporaneous if they are published **within the last four months**.
>
> Accordingly, we are not obligated to cite or compare to these works, and we believe this should not be viewed as a drawback of our submission. Notably, our work represents the pioneering exploration of academic presentation video generation. We would be happy to acknowledge these concurrent developments in a future revision.

---

> ### Author Response · Authors · 2025-11-21
> **Response (Part 2)**
>
> **W3**: *Beamer Might Limit Slide Stylization*
>
> We want to emphasize that **Beamer provides sufficient and comparable stylization** when compared to traditionally used PPT formats.
>
> ### (i) Slide customization in slide builder
> Moreover, our Slides Builder already provides substantial customization capabilities beyond standard templates. Specifically, users can:
>
> 1. **Customize visual appearance.** Users can specify explicit color-style prompts to achieve tailored color schemes and overall visual tone.
> 2. **Select composition layout.** The system supports choosing layout structures such as single-column or double-column typesetting.
> 3. **Incorporate custom icons.** Users may include institution or company logos by placing image files in the LaTeX project and prompting the system to integrate them into the slides.
> 4. **Control narrative structure.** The slide organization can be directed by specifying the desired sequence of sections (e.g., Motivation → Related Work → Method).
>
> These mechanisms allow users to generate slides with diverse structures and stylistic choices. Moreover, we plan to further enhance flexibility by supporting user-uploaded **Beamer templates**, which will offer more personalization in future versions.
>
> ### (ii) Beamer is suitable for academic use
> While Beamer naturally imposes certain stylistic conventions, it is highly suitable for academic presentations due to its formal, clear, and well-structured design. Moreover, within academic settings, Beamer is often perceived as more **professional** compared with traditional PPT-style formats.
>
> **Q1**: *Increase the Human Participants for Evaluation*
>
> Thank you for your suggestion. We **have added 15 additional participants**, resulting in a total of **25**. Since the primary audience of academic presentation videos is the research community, all participants are **graduate students with diverse backgrounds** (e.g., engineering, social sciences) to ensure the reliability. The instructions are shown below:
> > You will be given five academic paper presentation videos and asked to provide an overall ranking for these videos.
> When ranking, please consider the following four aspects: **Accuracy** (How complete, correct and reliable the content is considering that it represents a paper); **Fluency** (How coherent and smooth the narration is); **Visual Quality** (How clear and visually appealing the video is); **Audio Naturalness** (How natural and pleasant the audio sounds). Your ranking should reflect your overall impression across these four dimensions. To ensure objective evaluation:
> 1 Please watch all four videos carefully before ranking.
> 2 Base your decision solely on what you see and hear in the videos.
> 3 Evaluate each video independently before comparing them.
> 4 Avoid personal biases unrelated to the video content.
> Your responses are anonymous and will be used only for academic research.
>
> The results are shown below:
>
> | Human | PaperTalker | PresentAgent | VEO3 | Wan2.2 |
> |--------------------|----------------------------|-----------------------------|---------------------|------------------------|
> | 4.32 ± 0.79          | 3.76 ± 1.21               | 3.08 ± 0.56                 | 2.84 ± 1.12         | 1.00 ± 0.00            |
>
> The paired t-test between Human and PaperTalker shows no statistically significant difference **(t = 1.62, p = 0.12)**. Although Human receives slightly higher mean scores, the p-value indicates that this gap can be attributed to random variability, suggesting that **PaperTalker’s performance is statistically comparable to Human annotators**. In contrast, the paired t-test between PaperTalker and PresentAgent reveals a significant difference **(t = 2.53, p = 0.018)**. PaperTalker consistently achieves higher scores, **indicating a reliable performance advantage over PresentAgent across the evaluated samples**.
>
>
> **Q2**: *Human-made Videos May Have Cursor in Figure 7*
>
> Thank you for pointing this out. We will update it in future versions. However, in practice, many popular recording tools (e.g., Microsoft Office) still do not support recording presentation videos with a visible cursor, and such examples are also **rare on platforms like YouTube or SlidesLive**. This limitation also motivated us to explicitly include cursor rendering as a feature in our system.

---

> ### Author Response · Authors · 2025-11-21
> **Response (Part 3)**
>
> **Q3**: *VideoLLM Models Might Have Perferences During Evaluation*
> > Two of the three automatic evaluations use (video) LLMs as the viewer/judge, which can couple the evaluation distribution with model preferences; consider adding larger-scale human evaluation or user-behavior metrics.
>
> ### (i) Few VideoLLM models support multi-video inputs
> Not all existing multimodal models support **multi-video inputs with synchronized audio**, which fundamentally restricts the VideoLLM choice in evaluation. As shown in the table below, only a subset of models can process combined video–audio streams, and only Gemini-2.5 supports multiple videos as input:
>
> | Model               | Audio Input | Video Input | Video+Audio Combined | Multi-Video Input |
> |---------------------|-------------|-------------|-----------------------|--------------------|
> | **GPT-4o**          | ✔️ Yes       | ✔️ Yes       | ❌ No                 | ❌ No |
> | **GPT-4.1 Omni**    | ✔️ Yes       | ❌ No        | ❌ No                 | ❌ No |
> | **Qwen3 Omni**      | ✔️ Yes       | ✔️ Yes       | ✔️ Yes                | ❌ No |
> | **GLM-4.5V**        | ✔️ Yes       | ✔️ Yes       | ✔️ Yes                | ❌ No |
> | **Gemini-2.5-Pro**  | ✔️ Yes       | ✔️ Yes       | ✔️ Yes                | ✔️ Yes |
>
> This limitation reduces the range of models that can be fairly included in a multi-video, audio-aware evaluation protocol. Thus, it **prevents us from leveraging multiple models in a unified evaluation setting**, which would otherwise help mitigate potential model-specific preferences in the assessment.
>
> ### (ii) We add a larger-scale human evaluation.
> To address the concern, we additionally provide detailed human evaluations, as presented in **Q1**.

---

> ### Author Response · Authors · 2025-11-25
>
> Dear Reviewer cGYF,
>
> Thank you once again for your feedback!
>
> We would greatly appreciate it if you could review our response to ensure it adequately addresses your concerns. We remain fully dedicated to clarifying any remaining points and would welcome any further discussion to ensure all your questions are thoroughly answered.
>
> Thank you for your time and consideration.
>
> Best,
>
> Authors of Paper2Video

---

> > ### Comment · Reviewer_cGYF · 2025-11-28
> >
> > Thank you for your response. Nevertheless, given the limitations in the experimental validation, I still find the paper to be borderline.

---

> ### Author Response · Authors · 2025-12-01
>
> Thank you for the follow-up. We would like to offer two key clarifications regarding the experimental validation according to the questions in the official review.
>
> 1. **Scaled-up human evaluation further confirms the same finding.**
> Following your concern, we expanded our human evaluation from previous 10 to **25 graduate-level participants**, matching the target audience of academic presentation videos. The resulting statistical analysis shows that our method is **comparable to human performance** (non-significant difference in paired t-test **p = 0.12>0.05**) and **significantly outperforms** the baseline PresentAgent (significant difference in paired t-test **p = 0.018<0.05**), demonstrating the same finding.
>
> 2. **Compared baselines are comprehensive: end-to-end, multi-agent, and human**
>     1. The compared baselines in our experimental evaluation are comprehensive, encompassing **end-to-end methods** (Veo3, Wan2.2), **multi-agent frameworks** (PresentAgent, PPTAgent), and **human presenters**. We thoroughly evaluate each method across **slides**, **speech**, and **video** in terms of **information delivery** and **audience memory**. Notably, the three works mentioned by the reviewer all fall under only one category—multi-agent frameworks.
>     2. To the best of our knowledge, our work is the **first** to address academic presentation video generation. The three concurrent systems the reviewer mentioned all appeared **within two months of the ICLR 2026 submission deadline** (with one also under submission to ICLR 2026). As such, we are **not expected or required** to cite or compare against works that emerged post-submission. We therefore believe this **should not be considered a limitation** of our paper and we would be happy to acknowledge these concurrent developments in a future revision.
>
>
> We hope this clarifies the adequacy of our evaluation.

---

### Official Review · Reviewer_fecP · 2025-10-31

**Soundness:** 3
**Presentation:** 2
**Contribution:** 3
**Rating:** 8
**Confidence:** 3

**Summary:**

The Paper2Video framework offers an advancement in the automatic generation of academic presentation videos. By integrating multiple agents for slide creation, speech synthesis, and visual grounding, it addresses challenges of multi-modal coordination. The Paper2Video benchmark, containing 101 papers and corresponding videos, serves as a comprehensive tool for evaluating video quality, knowledge coverage, and memorability. The proposed system outperforms existing methods, demonstrating high alignment with human-made presentations.

**Strengths:**

1. The task of transforming academic papers into presentation videos is innovative and holds substantial future potential.
2. The final results demonstrate high effectiveness, with supplementary materials showcasing impressive video outputs.
3. The paper provides a detailed and clear description of the methodology, ensuring transparency and reproducibility.

**Weaknesses:**

1. **Lack of Novelty in Tree Search Visual Choice**: The PaperTalker framework appears to primarily integrate existing techniques. Specifically, the Tree Search Visual Choice lacks significant novelty, as similar approaches for layout optimization have been explored in other domains. A more innovative direction could involve designing a mechanism to generate figures on each slide that align better with the content being presented, offering deeper insights into how figures are tailored for more effective explanations.

2. **Incomplete Data in Table 2**: Table 2 contains numerous missing data points, which impacts the overall clarity and interpretability of the results. The absence of these data points could be addressed by either providing an explanation for the missing values or by utilizing alternative visual representations (e.g., bar charts or heatmaps) to present the missing information more effectively.

3. **Insufficient Dataset Size**: The 101 paper-video pairs used for evaluation may be insufficient to rigorously assess the method's generalizability. Given that large conferences often feature thousands of papers and videos each year, expanding the dataset would provide a more robust foundation for evaluating the model's performance across diverse topics and presentation styles.

**Questions:**

Can the PaperTalker framework handle papers from highly specialized domains? While the paper shows promising results in generating academic presentation videos, how well do you think PaperTalker will perform with highly specialized or niche academic papers, particularly in domains with complex, less standardized language or intricate visuals (e.g., advanced mathematical models or domain-specific terminologies)?

---

> ### Author Response · Authors · 2025-11-21
> **Response (Part 1)**
>
> Thank you for recognising the value of our work and for providing constructive and thorough feedback! We value your insights and hope our response addresses your comments.
>
> **W1**: *The Novelty in Tree Search Visual Choice*
> > The PaperTalker framework appears to primarily integrate existing techniques. Specifically, the Tree Search Visual Choice lacks significant novelty, as similar approaches for layout optimisation have been explored in other domains.
>
> Firstly, we highlight that **layout generation** is a fundamental yet highly practical problem, spanning tasks such as scene arrangement, object and poster composition, and slide layout design. While tree search has been explored in textual and agentic settings, **its efficient application to visual layout remains** underexplored and non-trivial. Our work is one of the pioneers addressing this gap, demonstrating the effective adaptation of tree-search strategies for visual layout refinement.
> ### (i) Necessity of tree search
> Compared with the **iterative slide-refinement pipelines** in previous slide generation method(e.g., PresentAgent), which require m rounds of refinement for each of the n slides (resulting in O(nm) time complexity), our method performs **only a single inference for each slide that contains layout issues**, reducing the complexity to O(n). This linear scaling makes Tree-Search Visual Choice substantially more efficient and suitable for large-scale presentation generation.
>
> Moreover, as shown in Table 5, our method achieves significant improvements in slide quality. Our approach **decouples parameter adjustment from optimal candidate selection**, circumventing the limitations of VLMs when these processes are intertwined. This separation leads to **more stable and reliable layout refinement**. Additional qualitative results are provided in **Figure 8**.
> ### (ii) Different tree search implementations
> To illustrate that applying tree-search methods to visual layout refinement remains non-trivial, we outline several potential strategies for integrating tree search into this problem:
>
> 1. **Individual Choice Scoring**
>    Each layout candidate is rendered into an image. The LLM scores each image independently, and the layout with the highest score is selected.
> 2. **Concat Visual Choice (Ours)**
>    All layout candidates are rendered and concatenated into a single composite image. The LLM views all candidates simultaneously and directly chooses the best layout, enabling more consistent global comparisons.
>
> Suppose we need to test \(k\) layout candidates. Here, **P** denotes the token cost of the prompt, and **L** denotes the token cost of the rendered slide image. We compare both implementations in terms of token cost and time cost. Tree Search Visual Choice achieves the lowest cost.
>
> | Method                   | Token Cost    | Time Cost              |
> |--------------------------|---------------|-------------------------|
> | Individual Choice Scoring           | k*(P + L)  | O(K) |
> | Concat Visual Choice (Ours)  | \(P + L\)     | O(1)       |
>
> **W2**: *Why There is Incomplete Data in Table 2*
>
> Thank you for raising this point. The original version did not include these results because **the choice of LLMs and VLMs primarily affects the slide and subtitle generation stage**, as these models are not used in the subsequent components (e.g., cursor generation and presenter synthesis). However, the generated outputs, such as slides and cursor prompts, may slightly influence subsequent stages of the pipeline. To provide a more complete experimental analysis, we have now included the corresponding results, which are reported in **Table 2**. All revisions in the manuscript are highlighted in red.
> | Method                    | Speech Sim. ↑ | Content Sim. ↑ | Arena ↑ | PresentQuiz Detail ↑ | PresentQuiz Under. ↑ | IP Memory ↑ | Avg. Duration (s) |
> |--------------------------|----------------|-----------------|---------|-----------------------|------------------------|--------------|--------------------|
> | PresentAgent_QWEN        | 0.045          | 0.24            | 1.2%    | 0.404                | 0.625                 | 10.4%            | 494.06                  |
> | PaperTalk_QWEN           | 0.646          | 1.66            | 10.8%   | 0.635                | 0.863                 | 39.5%            | 190.05                  |

---

> ### Author Response · Authors · 2025-11-21
> **Response (Part 2)**
>
> **W3**: *Small Benchmark Size*
>
> In terms of dataset size and robustness, we want to mention that Paper2Video is a **long-horizon agentic task**, introducing substantial challenges and distinctions compared to prior VLM benchmarks. While our benchmark contains 101 samples, each sample in Paper2Video is much heavier than instances in standard VLM task benchmarks. Below is a comparison of Paper2Video (101 samples) with a popular VLM task, MM-Vet [1] (218 samples):
>
> |                                 | **VLM task (MM-Vet)**            | **Agentic task (Paper2Video)**     |
> |---------------------------------|----------------------------------|--------------------------------------|
> | **Input**                       | Image + Question                 | Paper + Image + Audio                                |
> | **Input context length (avg.)** | 1 image + <1k tokens             |  > 22 images > 12k tokens            |
> | **Output length (avg.)**        | <100 tokens                      |  > 4mins video > 5,000 image > 4.6 million tokens         |
> | **Per sample time**             | Within seconds (single VLM gen.) | > 40 mins             |
> | **Number of Samples**           | 218                              | 101                                  |
>
> Given these increases in context, output, and runtime, scaling Paper2Video is far more resource-intensive than scaling existing VLM benchmarks. Our 101 carefully constructed samples already provide **broad coverage while keeping evaluation feasible and reproducible**.
>
> By contrast, MM-Vet can be completed within **minutes** (~a few seconds per item × 218 items), whereas a full Paper2Video run requires  **>40 hours** per model (>40 min per instance × 101 instances). Our current scale, therefore, strikes a pragmatic balance: large enough to expose systematic trends, yet light enough to encourage broad adoption.
>
> However, we greatly appreciate your viewpoint, and we will continue to expand the benchmark and release additional samples as resources permit in future work.
>
> [1] MM-Vet: Evaluating Large Multimodal Models for Integrated Capabilities. ICML 2024
>
> **Q1**: *Generalization for Different Disciplines*
>
> ### (i) We begin with AI conferences mainly for two reasons:
>
> 1. **AI conferences uniquely provide high-quality author-recorded presentations.**
> **Curating aligned paper–video pairs is non-trivial**. Outside of AI, many fields (e.g., Physics, Sociology) rarely publish a dedicated presentation video per paper, which limits the feasibility of large-scale collection. Even within AI conferences, we still face challenges such as incomplete metadata and inconsistent auxiliary materials; therefore, we manually filter papers with relatively complete information and supplement missing fields to maintain data reliability.
> 2. **AI conference paper already offer representativeness and diversity.**
> Our dataset spans **diverse AI subfields**, including CV, NLP, and general ML, and covers papers of varying lengths with **rich structures** (text, figures, tables, and mathematical content). These venues also enforce rigorous peer review, ensuring that the source papers are clear, well-organized, and of consistently high quality, making them a practical foundation for studying paper-to-video generation.
>
> Thus, the Paper2Video benchmark already includes **papers containing advanced mathematical models and domain-specific terminology**.
> For example, in our benchmark, Eilers et al. [2] introduce a generalized neural tangent kernel accompanied by **extensive mathematical formulation**. On this paper, PaperTalker attains accuracies of **0.92 (Detail) and 0.96 (Understanding)**, suggesting that the framework can perform reliably even when the content is technically demanding.
> Moreover, the use of LaTeX for slide generation provides precise and expressive support for mathematical notation, which makes the framework particularly capable for papers which involve equations, derivations, or symbolic reasoning.
>
> ### (ii) PaperTalker for extremely specialized domains:
> Nevertheless, the generation of slides fundamentally depends on **the LLM’s interpretation of the paper**. For highly specialized works whose concepts are not fully elaborated within the text or involve domain knowledge not explicitly introduced in the related literature, the LLM may still produce incomplete or imprecise interpretations. We consider enhancing the system with **retrieval or external knowledge access** to be a promising direction for improving robustness when dealing with domain-specific terminology or expert-level content.
>
> [2]. Eilers, Luke, Raoul-Martin Memmesheimer, and Sven Goedeke. "A generalized neural tangent kernel for surrogate gradient learning." Advances in Neural Information Processing Systems 37 (2024): 9026-9085.

---

### Comment · Area_Chair_FdUt · 2025-11-27
**Please review the authors' responses and provide feedback ASAP**

Dear Reviewers  fecP & cGYF,

Thank you for your essential contributions to the review process. The authors have submitted their responses to your initial reviews.

I kindly ask you to carefully review the authors' responses for this submission. Your timely assessment of how the authors have addressed your original concerns is a critical step in reaching a final decision.

Please provide your feedback and any necessary updates to your reviews as soon as possible to ensure we can meet our tight schedule for the discussion phase.

Your prompt attention to this matter is highly appreciated.

Regards,

-AC

---

### Author Response · Authors · 2025-12-01
**Thank AC and Summary of All Responses**

Dear AC,

We greatly appreciate reviewer's recognition of our approach as **innovative and well-designed** (fecP, DjH8) with **highly effective, publishable-quality presentation videos** (fecP, cGYF) and supported by **clear, reproducible methodology and comprehensive evaluation criteria** (fecP, DjH8).

* **Reviewer fecP** (score: 8) recognizes the strength and contributions of our work.
* **Reviewer cGYF** raises concerns primarily about the experimental validation: number of participants in human study(Q1) and compared baselines(Q2).
* **Reviewer DjH8** finds the contributions interesting, while still expressing concerns about dataset scale(Q3) and evaluation design (Q4).

We believe our responses have sufficiently addressed the concerns raised by each reviewer respectively. We would like to further emphasize four key clarifications:

#### Q1: **Scaled-up human evaluation further confirms the same finding.**
We have scaled up expanded our human evaluation from 10 to **25 participants** and provide in-depth analyse as required by reviewer cGYF and DjH8. The resulting statistical analysis shows that our method is **comparable to human performance** (non-significant difference in paired t-test **p = 0.12>0.05**) and **significantly outperforms** the baseline PresentAgent (significant difference in paired t-test **p = 0.018<0.05**), demonstrating the same finding.
#### Q2: **Compared baselines are comprehensive: end-to-end, multi-agent, and human.**
1. Our experimental evaluation compares a comprehensive set of baselines, including **end-to-end methods** (Veo3, Wan2.2), **multi-agent frameworks** (PresentAgent, PPTAgent), and **human presenters**. We systematically assess all methods across slides, speech, and videos in terms of information delivery and audience memory. We note that the three works referenced by the reviewer cGYF all belong to the multi-agent frameworks category.
2. To the best of our knowledge, our work is the **first** to study academic presentation video generation. The three concurrent systems referenced by the reviewer appeared **within two months** of the ICLR 2026 submission deadline. Therefore, these works fall **outside the scope** of required citation or comparison at submission time. Please also see our detailed comment to the AC under reviewer cGYF.
#### Q3: **Scale of Long-horizon Agentic tasks**.
Paper2Video is a long-horizon agentic task, where each instance about **4.6M tokens**, making large-scale expansion substantially more resource-intensive than conventional VLM benchmarks. Under this cost profile, **101 samples represent a large effective evaluation load**, and our scale is aligned with prior multimodal-generation agentic benchmarks such as Paper2Poster[1] (**100** samples), Paper2Code[2] (**90** samples) and PaperBench[3] (**20** samples).
#### Q4: **Our Evaluation is Comprehensive and Fairly Designed**.
To the best of our knowledge, we are the **first** to explicitly articulate **what constitutes a *“good”* academic presentation video** and to develop evaluation metrics grounded in how human viewers actually process such content. Specifically, we evaluate **(i)** slide and speech quality, **(ii)** the effectiveness of information delivery, and **(iii)** audience academic memory. These complementary perspectives provide a holistic assessment of how well a generated presentation functions both visually and communicatively. In addition, we explicitly **control for sequential bias** to ensure fairness and reliability in VideoLLM evaluation.

Thank you for your time and effort in reviewing our paper. We sincerely appreciate your consideration.

Authors of Paper2Video

[1] Pang, Wei, et al. “Paper2Poster: Towards Multimodal Poster Automation from Scientific Papers.” Advances in Neural Information Processing Systems (2025).

[2] Seo, Minju, et al. "Paper2code: Automating code generation from scientific papers in machine learning." arXiv preprint arXiv:2504.17192 (2025).

[3] Starace, Giulio, et al. "PaperBench: Evaluating AI’s Ability to Replicate AI Research." Forty-second International Conference on Machine Learning.

---

### Meta-Review · Area_Chair_P2ue · 2025-12-27

**Summary:**

The initial reviews acknowledged the practical value and interest of the proposed "PaperTalker" system but raised significant concerns regarding the dataset scale, evaluation methodology, and technical novelty. While the authors provided a detailed rebuttal—including expanding the human evaluation and explaining the computational constraints of the benchmark—the consensus among the critical reviewers remains that the work is limited by its experimental validation. Furthermore, the core methodology appears to be a combination of existing tools rather than a fundamental scientific innovation. Given that the concerns regarding the robustness of the benchmark and the "engineering" nature of the contribution remain outstanding, I recommend Rejection.

**Reviewer Concerns:**

Addressed during Rebuttal:
- Missing Data & Baselines: The authors successfully filled in missing data in Table 2 and justified the exclusion of very recent (contemporaneous) baselines based on conference policy.
- Input Consistency: The rebuttal clarified how input conditions (paper length, templates) were controlled during comparisons with baselines like PresentAgent and Veo3.
- Style Flexibility: The authors clarified the customization capabilities within the Beamer-based pipeline.

Outstanding/Minor Points:

- Scientific Novelty vs. Engineering: Reviewer fecP’s concern that the framework primarily integrates existing techniques (Tree Search, LLMs) was not fully alleviated. The method is viewed more as an engineering application of established tools rather than a methodological breakthrough.

- Benchmark Scale & Validity: Both reviewers cGYF and DjH8 remained unconvinced by the dataset size ($N=101$). While the authors explained that the "agentic" nature of the task makes scaling expensive, this does not mitigate the reviewers' concerns regarding the statistical power and generalizability of the benchmark.

- Evaluation Robustness: Despite increasing the user study participants from 10 to 25, Reviewer DjH8 maintained that the sample size is small and the evaluation design (relying heavily on custom LLM-based metrics like PresentArena) lacks sufficient bias control and rigorous validation.

- Cursor Generation: The assumption of sentence-level cursor movement was criticized as unnatural compared to human behavior, a limitation the authors acknowledged but could not fully resolve in the current scope.

**Reviewer Scores:**

Reviewer fecP (Rating: 8 $\to$ 6): While this reviewer was initially positive about the application, they noted the "Lack of Novelty" as a weakness. Had they participated in the discussion regarding the benchmark's limitations and the engineering nature of the solution, they would likely have lowered their score to align with the consensus on technical depth.

Reviewer cGYF (Rating: 4 $\to$ 4): The reviewer explicitly stated after the rebuttal that despite the clarifications, the limitations in experimental validation and insufficient data scale prevented them from raising the score.

Reviewer DjH8 (Rating: 4 $\to$ 4): This reviewer appreciated the additional analysis but explicitly maintained their score. They concluded that while the contribution is interesting, it is fundamentally limited by the current evaluation design and the small scale of the dataset.

---

### Decision · Program_Chairs · 2026-01-26

Reject